# Deep mutational learning for the selection of therapeutic antibodies resistant to the evolution of Omicron variants of SARS-CoV-2

Lester Frei [1,2,5], Beichen Gao [1,2,5], Jiami Han [1,2], Joseph M. Taft [1,2], Edward B. Irvine [1], Cédric R. Weber [3], Rachita K. Kumar [1], Benedikt N. Eisinger [1], Andrey Ignatov [1], Zhouya Yang[1] & Sai T. Reddy [1,2,4] ✉

Most antibodies for treating COVID-19 rely on binding the receptor-binding domain (RBD) of SARS-CoV-2 (severe acute respiratory syndrome coronavirus 2). However, Omicron and its sub-lineages, as well as other heavily mutated variants, have rendered many neutralizing antibodies ineffective. Here we show that antibodies with enhanced resistance to the evolution of SARS-CoV-2 can be identified via deep mutational learning. We constructed a library of full-length RBDs of Omicron BA.1 with high mutational distance and screened it for binding to the angiotensin-converting-enzyme-2 receptor and to neutralizing antibodies. After deep-sequencing the library, we used the data to train ensemble deep-learning models for the prediction of the binding and escape of a panel of eight therapeutic antibody candidates targeting a diverse range of RBD epitopes. By using in silico evolution to assess antibody breadth via the prediction of the binding and escape of the antibodies to millions of Omicron sequences, we found combinations of two antibodies with enhanced and complementary resistance to viral evolution. Deep learning may enable the development of therapeutic antibodies that remain effective against future SARS-CoV-2 variants.

The onset of the coronavirus disease 2019 (COVID-19) pandemic spurred the rapid discovery, development and clinical approval of several antibody therapies. The monoclonal antibody LY-CoV555 (bamlanavimab) (Eli Lilly)[1] and the combination therapy consisting of REGN10933 (casirivimab) and REGN10987 (imdevimab) (Regeneron)[2] were among the first to receive Emergency Use Authorization (EUA) from the US Food and Drug Administration in late 2020. The primary mechanism of action for these therapies consists of virus neutralization by binding to specific epitopes of the receptor-binding domain (RBD) of SARS-CoV-2 (severe acute respiratory syndrome coronavirus 2) spike (S) protein, thus inhibiting viral entry into host cells via the angiotensin-converting enzyme 2 (ACE2) receptor. However, the emergence of SARS-CoV-2 variants such as Beta, Gamma and Delta, each characterized by numerous mutations in the RBD, showed reduced sensitivity to neutralizing antibodies, including LY-CoV555[3,4], the EUA of which was subsequently revoked. It is worth noting that antibody combination therapies such as those from Regeneron and Eli Lilly (LY-CoV555+LY-CoV16 (etesevimab)) showed higher resilience to viral variants and maintained their EUA throughout most of 2021[3]. However, the emergence and rapid spread of Omicron BA.1 in late 2021, a variant which has a staggering 35 mutations in the S protein, 15 of which are in the RBD, resulted in substantial escape from nearly all clinically approved antibody therapies[5]. This includes the combination therapies from Regeneron and Eli Lilly, which also had their EUAs subsequently

[1]Department of Biosystems Science and Engineering, ETH Zurich, Basel, Switzerland. [2]Basel Research Centre for Child Health, Basel, Switzerland. [3]Alloy Therapeutics (Switzerland) AG, Allschwil, Switzerland. [4]Botnar Institute of Immune Engineering, Basel, Switzerland. [5]These authors contributed equally: Lester Frei, Beichen Gao. ✉e-mail: sai.reddy@ethz.ch

revoked[6]. Even antibody therapies with exceptional breadth, which were initially discovered against the ancestral SARS-CoV-2 (Wu-Hu-1) and retained neutralizing activity against BA.1—S309 (sotrovimab) (GSK/Vir)[7] and LY-CoV1404 (bebtelovimab) (Eli Lilly)[8]—lost efficacy against subsequent Omicron sublineages (such as BA.2, BA.4/5 and BQ.1.1)[9,10] and had their clinical use authorization revoked. Thus, all previously approved antibody therapies are no longer authorized for clinical use, despite their critical need for the protection of at-risk populations (young children, the elderly, individuals with chronic illnesses and those with weakened immune systems)[11–15].

The ephemeral clinical life span of COVID-19 antibody therapies has emphasized that, in addition to established metrics for antibody therapeutics (for example, neutralization potency, affinity and developability)[16], it is imperative to evaluate antibody breadth (ability of an antibody to bind to divergent SARS-CoV-2 variants) at an early stage of clinical development. This may enable selection and focused development of lead candidates that have the most potential to maintain activity against a rapidly mutating SARS-CoV-2, such as ZCB11—a broadly neutralizing antibody that maintains neutralization to Omicron variants and was discovered from memory B cells of an mRNA (BNT162b2) vaccinated donor[17]. To address this, high-throughput protein engineering techniques such as deep mutational scanning (DMS)[18] have been extensively used to profile the impact of single position mutations in the RBD on ACE2-binding and antibody escape[5,19–24]. While DMS has proven effective for profiling single mutations, many SARS-CoV-2 variants that have emerged possess multiple mutations in the RBD. For example, the aforementioned Omicron BA.1 lineage or the more recently identified BA.2.86, which possesses an astonishing 13 RBD mutations relative to its closest Omicron variant (BA.2) and 26 RBD mutations relative to ancestral Wu-Hu-1[25–27]. Experimental screening of combinatorial RBD mutagenesis libraries using display platforms such as yeast surface display vastly undersamples the theoretical protein sequence space, and therefore computational approaches are increasingly being used in concert. For instance, experimental measurements such as DMS data have been used to calculate statistical estimators[28] or to train machine learning models that make predictions on ACE2 binding and antibody escape[29–31]. While such computational tools enable interrogation of a larger mutational landscape of SARS-CoV-2, their primary reliance on datasets that largely consist of single mutations from DMS experiments limits their ability to capture epistatic effects of combinatorial mutations, especially in the context of high-mutational variants such as Omicron sublineages (for example, BA.1, BA.4/5, BA.2.86).

Here we apply deep mutational learning (DML), which combines yeast display screening, deep sequencing and machine learning to address the emergence of Omicron BA.1 and its many sublineages. We expand the scope of DML from screening short, focused mutagenesis libraries[32] to screening combinatorial libraries spanning the entire RBD for binding/escape to ACE2 or antibodies (Fig. 1a). Ensemble deep learning models using dilated residual network blocks were trained with deep sequencing data and shown to make accurate predictions for ACE2 binding and antibody escape. Next, deep learning was used to determine the breadth of second-generation antibodies (with known binding to BA.1) across a massive sequence landscape of BA.1-derived synthetic lineages, allowing the rational selection of specific antibody combinations that optimally cover the RBD mutational sequence space (Fig. 1b). This approach provides a powerful tool to guide the selection of antibody therapies that have enhanced resistance to both current and future high-mutational variants of SARS-CoV-2.

## Results

### Design and construction of a high-distance Omicron BA.1 RBD library

A mutagenesis library was constructed based on BA.1, covering the entire 201 amino acid RBD region (positions 331–531 of SARS-CoV-2 S protein). To maximize the interrogated RBD sequence space, the library design was entirely synthetic and unbiased, as it did not consider evolutionary data or previous experimental findings. For the construction of the library, the RBD sequence was split into 11–12 fragments, each with an approximate length of 48 nucleotides (Supplementary Table 1). For a fragment of average length, 137 different single-stranded oligonucleotides (ssODN) were designed, where each ssODN had either zero, one or all possible combinations of two codons replaced by fully degenerate NNK codons (N = A, G, C or T; K = G or T) (Fig. 2a and Methods). In total, 6,298 ssODNs were used to construct the library. For each fragment, ssODNs were amplified using PCR to generate double-stranded DNA. Each fragment was flanked by recognition sites for the type II-S restriction enzyme BsmBI, thus enabling assembly into full-length RBD regions using Golden Gate assembly (GGA)[33]. GGA uses type II-S restriction enzymes capable of cleaving DNA outside their recognition sequence, thereby allowing the resulting DNA overhangs to have any sequence. Based on the overhangs, individual fragments were assembled by DNA ligase to full-length RBD sequences with high fidelity[34,35]. The restriction sites were eliminated during the process, thus enabling scarless assembly of full-length RBD sequences (Fig. 2b and Methods)[34]. This approach yielded approximately 98% correctly assembled RBD sequences (Supplementary Fig. 1). As GGA required four nucleotide homology between individual fragments for ligation, this led to portions of the sequence which needed to remain constant, thereby restricting library diversity[36]. To overcome this limitation, four staggered sub-libraries were designed and individually assembled. Using sub-library 1 as a reference, sub-library 2 is shifted by 12 nucleotides, sub-library 3 by 24 nucleotides and sub-library 4 by 36 nucleotides. These sub-libraries provided an increase in the mutational space covered by the RBD combinatorial mutagenesis library, as at the GGA homology region for a given library, the remaining three libraries can have mutations (Fig. 2c). Considering all possibilities of combining fragments with either zero, one or two mutations, this design led to a theoretical library diversity of approximately $10^{42}$ variants.

The current read length of Illumina does not allow coverage of the entire RBD with a single sequencing read (paired end). Therefore, two separate sequencing libraries (seq-libraries A and B) were individually constructed. Seq-libraries A and B possessed mutations in positions 331–475 and 386–531, respectively (Fig. 2c). The seq-libraries were constructed separately, but all subsequent steps were performed in a pooled fashion. For targeted sequencing, each seq-library was flanked by unique primer binding sites. Following deep sequencing, complete mutational coverage for each residue was observed in both seq-libraries (Fig. 2d). It is worth noting that the mutational frequency is somewhat variable across the seq-libraries, showing a marked decrease in mutations every 16 residues. The low mutational frequencies line up with GGA homologies of sub-library 1. We hypothesize that when pooling the sub-libraries, sub-library 1 was more prominent than the other sub-libraries, and therefore less mutations at these sites are observed.

Next, to optimize the number of mutations per RBD sequence, a titration of the fragment assembly step was performed. Wild-type fragments (BA.1 sequence) and fragments with one and two mutations, respectively, were pooled in different ratios for assembly. Separately, assembly was performed with 60%, 70% and 80% of wild-type fragments, with the remaining percentage split evenly between fragments with one and two mutations. Deep sequencing of these libraries revealed a clear trend in mutational distribution based on the different ratios, highlighting the tunable nature of our approach (Fig. 2e). Based on these results, all subsequent work was carried out using the 60% wild-type library as it has the highest mean number of mutations, therefore allowing us to adequately model and profile extensively mutated Omicron sublineages.

### Screening RBD libraries for ACE2 binding and antibody escape

Co-transformation of yeast cells (*Saccharomyces cerevisiae*, strain EBY100) using the PCR-amplified RBD library and linearized plasmid

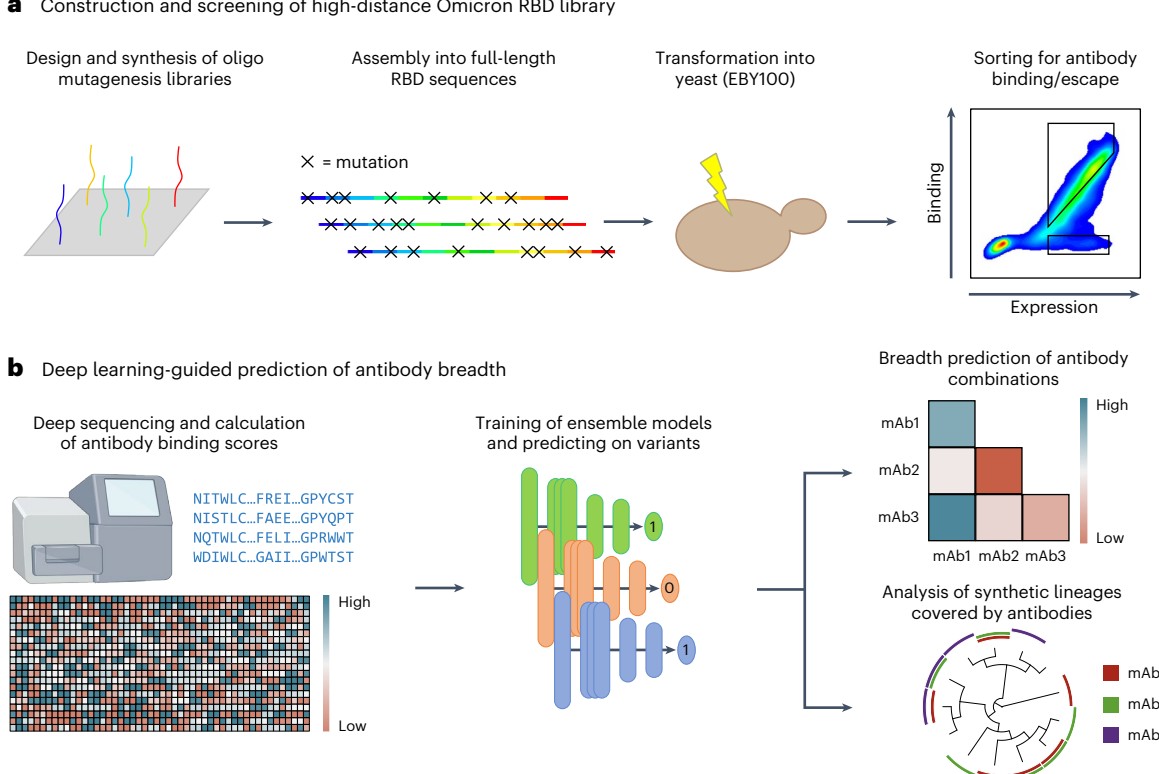

**Fig. 1 | Overview of DML to guide the selection of antibodies targeting SARS-CoV-2 RBD. a**, For the construction of the high-distance Omicron BA.1 RBD library, short ssODNs are designed to possess mutations. The fragments are assembled into full-length RBD sequences using GGA and transformed into yeast. The resulting library is screened for antibody binding and escape using FACS.

**b**, The sorted high-distance RBD variant library is deep sequenced, and the data are used to train ensemble deep learning models to predict ACE2 and antibody binding or non-binding (escape). Deep learning models are used to predict the breadth of antibody combinations as well as their binding to synthetic RBD variants and lineages. mAb, monoclonal antibody.

yielded more than $2 \times 10^8$ transformants (Methods). Yeast surface display of RBD variants was achieved through C-terminal fusion to Aga2[37]. Next, fluorescence-activated cell sorting (FACS) was used to isolate yeast cells expressing RBD variants that either retained binding or completely lost binding to dimeric soluble human ACE2 (Fig. 3a). It is worth noting that RBD variants with only partial binding to ACE2 were not isolated, as such intermediate populations could not be confidently classified as either binding or non-binding. Removing these variants is essential to obtain cleanly labelled datasets for training supervised machine learning models. As binding to ACE2 is a prerequisite for cell entry and subsequent viral replication, only this population is biologically relevant. Thus, only the ACE2-binding population was used in subsequent FACS steps to isolate RBD variants that either retained binding or completely lost binding (escape) activity to a panel of eight neutralizing antibodies (Fig. 3a,b, Supplementary Fig. 2 and Supplementary Table 2). The antibodies selected target different epitopes and are well characterized for their neutralizing activity to BA.1 and its sublineages, which provide a good internal control to assess the accuracy of our method[38-40]. The panel consists of the following antibodies: A23-58.1 (ref. 41), COV2-2196 (ref. 42), Brii-198 (ref. 43), ZCB11 (ref. 17), 2–7 (ref. 44), S2X259 (ref. 45), ADG20 (ref. 46) and S2H97 (ref. 20).

Following ACE2 and monoclonal antibody sorting, pure populations of RBD variants (binding and non-binding) were subjected to deep sequencing (Supplementary Table 3). During the sorting process, it was noted that antibodies COV2-2196 and 2–7 show a weaker binding signal (Supplementary Fig. 2). This was especially pronounced in the case of antibody 2–7 and is likely due to the low affinity of this antibody to Omicron BA.1 RBD (Supplementary Fig. 3) and a generally low mutational resilience (Supplementary Fig. 4a). Those factors contributed to the collection of fewer cells for those antibodies. Reads covering the

RBD sequence were then extracted from the deep sequencing data, and heat maps were constructed depicting binding scores (relative amino acid frequencies per position in the RBD of binding versus non-binding variants) (Fig. 3c,d and Supplementary Figs. 4 and 5). The heat maps show nearly complete coverage of mutations across the RBD within all sorted populations. A heterogeneous distribution of mutations is observed for ACE2 binding, with no specific positions or mutations showing dominance (Fig. 3c). This agrees with previous studies that suggest the Q498R and N501Y mutations present in BA.1 show strong epistatic effects that compensate for many mutations that cause loss of binding[47]. By contrast, for certain antibodies, clear mutational patterns could be observed, including escape mutations that correspond with previous DMS studies (Fig. 3d,e and Supplementary Figs. 4 and 5). For example, RBD escape variants for Brii-198 are enriched for mutations in positions 346 and 452 (Supplementary Fig. 4d), which are present in BA.1 and BA.4/BA.5, respectively, and correspond to previous work that shows they drive a drastic loss of binding to Brii-198 (ref. 48). By contrast, enrichment of these escape mutations are not observed for antibody 2–7 (Supplementary Figs. 4 and 5), even though Brii-198 and 2–7 share a similar epitope, suggesting that the binding modality between these two antibodies are different, which is also reflected by their difference in resistance to Omicron variants (for example, 2–7 shows strong binding to BA.2 and BA.4/BA.5, while Brii-198 does not bind BA.2.12 and BA.4/BA.5)[10,39]. Similarly, the F486V mutation, which has been demonstrated to drastically reduce the neutralization potency of ZCB11 by over 2,000-fold[10], is highly enriched in the RBD escape population (Fig. 3d,e). These mutations are also seen in A23-58.1 and COV2-2196, which bind to a similar epitope (Supplementary Figs. 4 and 5). Lastly, for ADG20, we observe a high enrichment of escape mutations in 408 (Fig. 3e and Supplementary Figs. 4 and

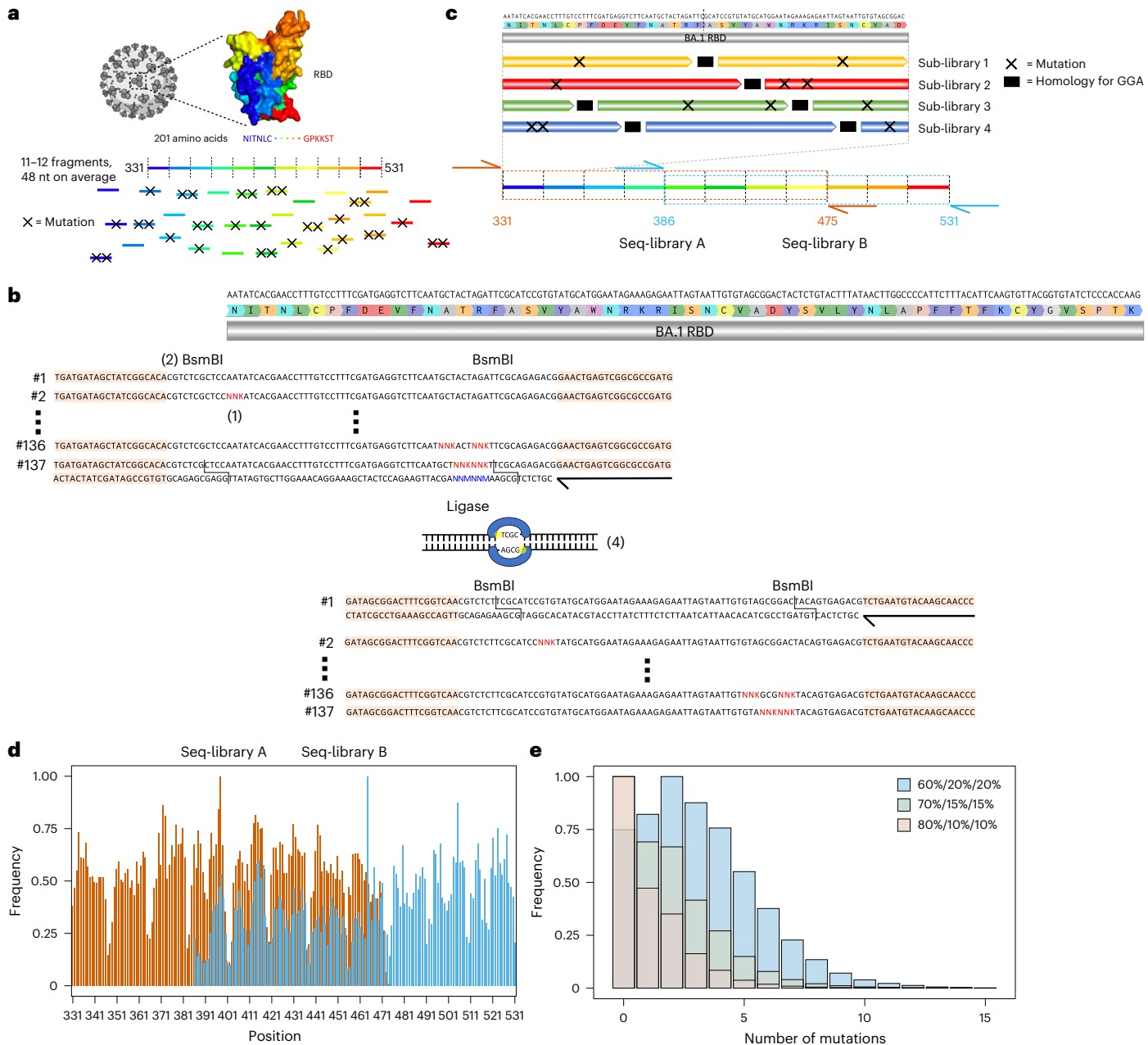

**Fig. 2 | Construction of a high ED synthetic variant library based on Omicron BA.1 RBD. a**, The RBD sequence was split into 11–12 fragments, each being on average 48 nucleotide in length. For each fragment, a ssODN library with either zero, one or two mutations was designed. **b**, To introduce mutations, NNK codons were tiled across the fragments (1). Each fragment was flanked by BsmBI sites (2). The ssODNs were flanked by primer binding sites for double-stranded synthesis through PCR (primers are represented by black arrows, and primer binding sites are peach coloured) (3). The type II-S restriction enzyme BsmBI gives rise to orthogonal four nucleotide overhangs, which are used by a ligase to assemble individual fragments into full-length RBD sequences (4). **c**, The use of GGA for library construction required the presence of constant regions for ligation between fragments (in black), thereby restricting the library diversity. To overcome this drawback, four staggered sub-libraries were constructed. Due to limitations in sequencing length, it was further necessary to split the RBD into two separate libraries. The extent of seq-library A is indicated in orange and seq-library B in cyan. The primer binding sites for deep sequencing are indicated using orange and cyan arrows. **d**, Targeted sequencing of seq-libraries A and B showed comprehensive mutational coverage for both libraries. The same colour scheme as in **c** was used to indicate the extent of both libraries. **e**, To adjust the mutational rate of the library, different ratios of fragments with zero, one or two mutations (60%/20%/20%, 70%/15%/15% and 80%/10%/10%) were pooled, yielding libraries with average number of mutations of 3.59, 2.07 and 1.41, respectively.

5); this position is also mutated in BA.2 and BA.4/BA.5 variants, which have been shown to have drastically reduced neutralization by ADG20 (ref. 10). While heat map analysis allows specific mutational patterns to be linked with antibody escape profiles, the high-dimensional nature—and potentially higher order impact—of combinatorial mutations is not reflected in this format. It is apparent that protein epistasis and combinatorial mutations can modify the effect of known escape mutations, either amplifying or reducing antibody binding. For example, individual RBD mutations (G339D, S371F, S373P, S375, K417N, N440K, G446S, S477N, T478K, E484A, Q493R, G496S, Q498R, N501Y, Y505H) in BA.1 and BA.1.1 do not enhance escape to COV2-2196, with each mutation causing an average fold reduction of 2.2, but together cause over 200-fold reduction in neutralization[49]. Conversely, the intro-duction of the single R493Q mutation in BA.2 substantially rescued the

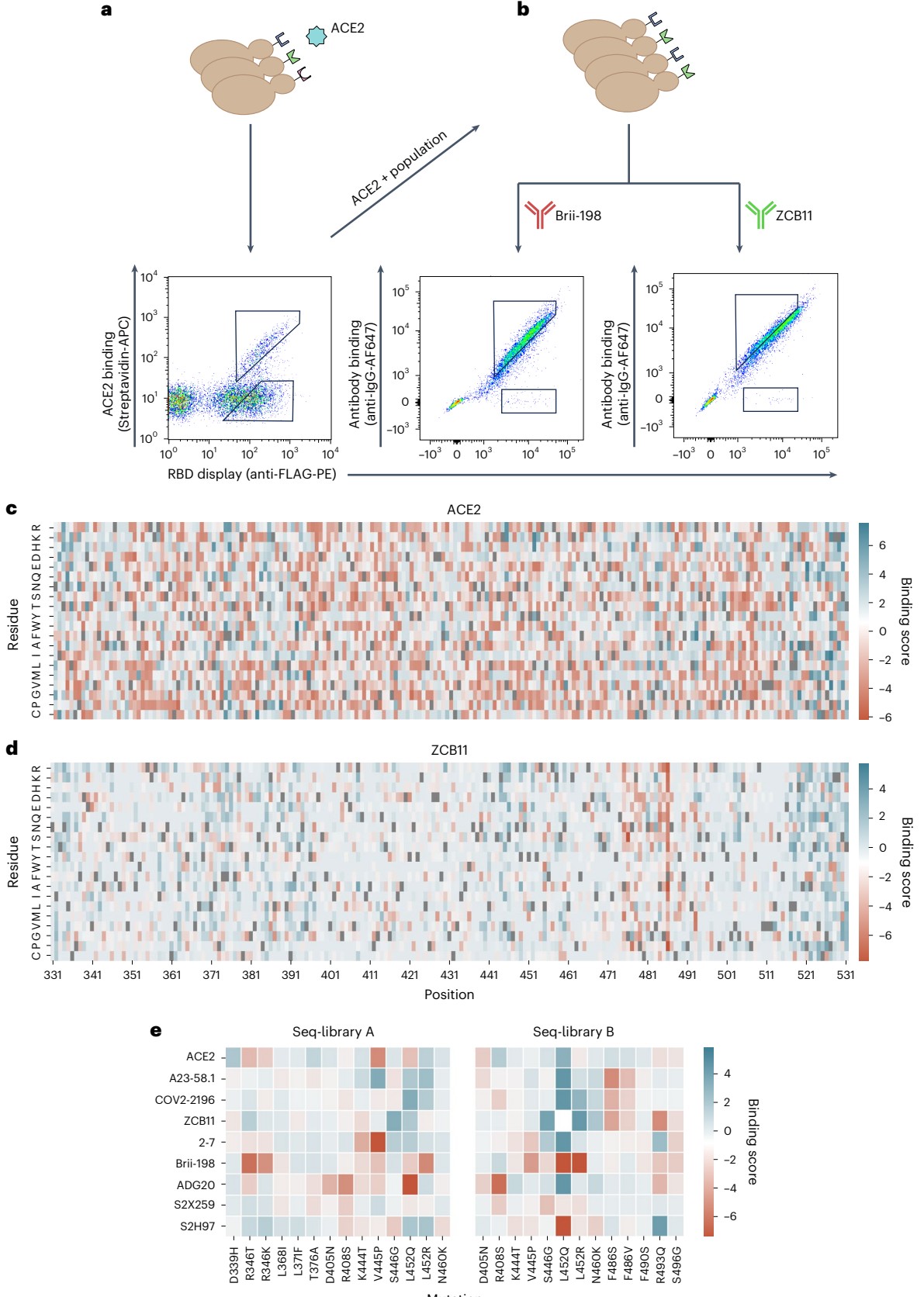

**Fig. 3 | Screening RBD libraries for ACE2 binding and antibody escape by yeast display and deep sequencing. a,b**, Workflow for sorting of yeast display RBD libraries and FACS dot plots for ACE2 (**a**) and antibodies Brii-198 and ZCB11 (**b**). Gating schemes correspond to binding and non-binding (escape) RBD variant populations. **c,d** Heat maps depicting the binding score of each amino acid per position of full-length RBD following sorting and deep sequencing of libraries for ACE2 (**c**) and ZCB11 (**d**); higher binding score indicates greater frequency in the binding population versus non-binding population. Wild-type BA.1 residues are in grey. **e**, Heat maps for seq-libraries A and B depicting binding scores for ACE2 and antibodies of key mutations seen in major Omicron sublineage variants.

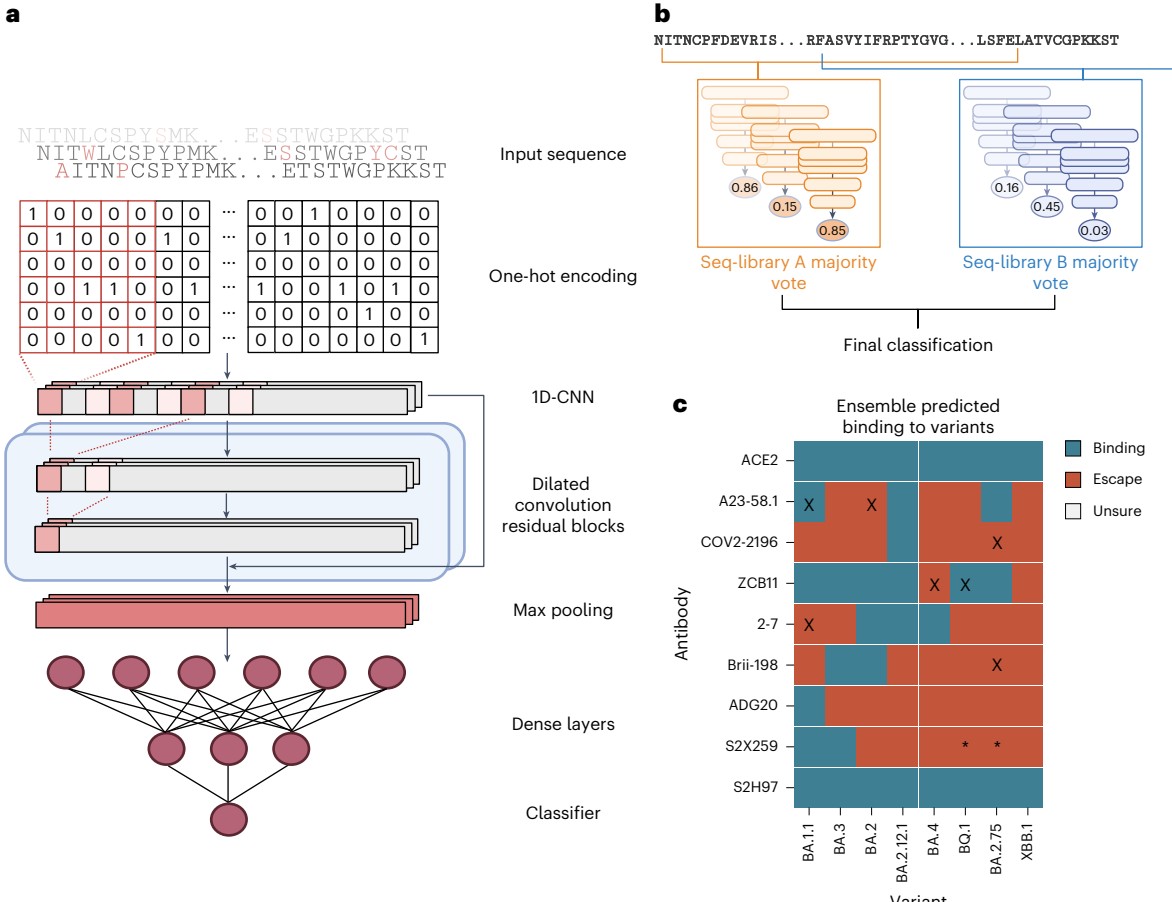

**Fig. 4 | Training and testing of deep learning ensemble models for prediction of ACE2 binding and antibody escape based on full-length RBD sequences.**
**a**, Deep sequencing data of sorted yeast display libraries are encoded by one-hot encoding and used to train CNN models with several dilated convolutional residual blocks. The models perform a final classification by predicting binding or non-binding to ACE2 or antibodies based on the encoded RBD sequence.
**b**, Majority voting by an ensemble of models is used to determine the final label for each variant. **c**, Predicted labels of antibodies to well-characterized Omicron variants; colours indicate final labels; mis-classifications are marked with an 'X'; conflicting data for S2H259 binding to BQ.1 and BA.2.75 are marked with '*'.

neutralizing activities of Brii-198, REGN10933, COV2-2196 and ZCB11 (ref. 10). Thus, while the heat maps indicate specific mutational contributions to antibody escape, other techniques such as deep learning are required to capture the high-dimensional nature of combinatorial mutations and generalize to future mutations.

## Deep learning ensemble models accurately predict ACE2 binding and antibody escape

To address the high dimensionality of our dataset and to understand epistatic effects between mutations in the full RBD mutational sequence space, which is far too vast to be comprehensively screened experimentally, we trained deep learning ensemble models. Deep sequencing data from FACS-isolated yeast populations underwent pre-processing and quality filtering before being used as training data for machine learning. In the datasets for all antibodies, using the BA.1 RBD sequence as a reference, the mean rate of mutations ranged between edit distance (ED) two ($ED_2$) and $ED_3$, with a max $ED_8$ (Supplementary Figs. 6 and 7 and Methods). Following DNA to protein translation, one-hot encoding was performed to convert amino acid sequences into an input matrix for machine and deep learning models (Fig. 4a). Supervised machine learning models were trained to predict the probability ($P$) that a specific RBD sequence will bind to ACE2 or a given antibody. A higher $P$ signifies a stronger correlation with binding, whereas a lower $P$ corresponds to non-binding (escape). The machine learning models tested included K-nearest neighbour, logistic regression, naive Bayes,

support vector machines and random forests. In addition, as a baseline for deep learning models, a multilayer perceptron (MLP) model was also tested. Finally, we implemented a convolutional neural network (CNN) inspired by ProtCNN[50], which leverages residual neural network blocks and dilated convolutions to learn global information across the full RBD sequence (Fig. 4a).

Each model was trained using an 80/10/10 train–validate–test split of data. Inputs were one-hot encoded RBD sequences, with the CNN using a two-dimensional (2D) matrix and others using a 1D flattened vector. For initial benchmarking, a collection of different baseline machine learning models, as well as CNN and MLP deep learning models, were trained on each dataset with hyperparameter optimization through random search and were evaluated with fivefold cross validation based on several common metrics (accuracy, measure of predictive performance (F1), Matthews correlation coefficient (MCC), precision and recall). During training, class balancing was achieved by upsampling the minority class in the training set, while the test set remained unbalanced. Comparing performances of the baseline models, both extreme gradient boosting (XGBoost) and CNN models obtained the highest MCC scores across most of the antibodies and libraries (Supplementary Data 1). However, in one single condition, seq-library A for antibody S2X259, the CNN model vastly outperformed all of the others, with an MCC score 0.15 higher than the next best model (Supplementary Data 1). This suggests that depending on the antibody, the use of deep learning architectures is still crucial for learning complex

interactions across larger distances, and thus we performed all subsequent work with this CNN architecture.

We next applied an exhaustive hyperparameter search on CNN models to optimize their performance (Supplementary Table 4). To prevent data leakage during training, the held-out test set was fixed, and multiple models were trained on different training–validation splits of the remaining dataset to make sure each model learned slightly different parameters of the data. When tested on the held-out test set, the final models yielded robust predictive performance up to an $ED_8$ from the wild-type BA.1 sequence (Supplementary Fig. 8 and Supplementary Table 5).

For our final ensemble, we selected three CNN models from each library with the highest MCC scores to generate the predicted labels for each RBD variant through majority voting (Fig. 4b). In short, each model outputs $P$ of binding for each input sequence, and labels are assigned based on a threshold. An RBD variant was assigned a predicted 'escape' label if the ensemble models of either seq-library A or seq-library B predicted escape, and assigned a predicted 'binding' label only if both ensemble models predicted binding. We tested the performance of the ensemble models using published experimental data of antibody binding (or neutralization) to Omicron sublineages[10,38,39,48,51–53] (Supplementary Data 2). Where possible, we used published antibody affinity data[39] and set the escape (non-binding) threshold to $K_d > 100$ nM, a limit that indicates considerable loss of binding. For some antibodies, such as ZCB11 and Brii-198, neutralization data are only reported without affinity measurements[10,48], and therefore in these cases, we used neutralization to define an escape threshold of half maximal inhibitory concentration ($IC_{50}$) $> 10$ μg ml$^{-1}$. For our deep learning models, standard thresholds were used for classification: $P > 0.5$ as binding, and $P \leq 0.50$ as non-binding (escape). The models assigned accurate labels for 87.5% (63/72) of ACE2-RBD variant or antibody-RBD variant pairs, with two false positives, and seven false negatives (Fig. 4c). The high number of false negatives suggests that the models tend to be more conservative for predictions of antibody binding—which may be preferable in the context of selecting optimal antibody therapies. Furthermore, 6 out of 9 misclassifications occur when variants have >8 mutations from the parental BA.1 sequence (BA.4, BA.2.75, BQ.1), which may suggest that at high mutational loads, the combinatorial effects of mutations begin to exceed the predictive threshold. Finally, there have been contradictory reports for binding and neutralization with the antibody S2X259, one publication reporting low half maximal effective concentration ($EC_{50}$) and $IC_{50}$ values across all variants up to BQ.1 (ref. 39); however, a second study[10] reports a much higher $IC_{50}$ of $>10$ μg ml$^{-1}$ to BA.2. Our model predictions support the latter publication, as we predict a loss of binding of S2X259 to all variants beyond BA.2.

### Designing antibody combinations by predicting resistance to synthetic Omicron lineages

After validating the performance of CNN models on test and validation data, we next deployed them to evaluate the resistance of antibodies to viral evolution. While antibody breadth is normally evaluated retroactively based on neutralization or binding to previously observed variants, here we aimed to leverage this machine-learning-guided protein engineering approach to prospectively characterize and assess the breadth of antibodies against Omicron variants that may emerge in the future. This was achieved by generating synthetic lineages

stemming from BA.1. As the potential sequence space of combinatorial RBD mutations is exceedingly massive, it was necessary to reduce this to a relevant subspace, and therefore mutational probabilities were calculated across the RBD using SARS-CoV-2 genome sequencing data (available on Global Initiative on Sharing Avian Influenza Data, GISAID (www.gisaid.org)) and used to generate synthetic lineages that mimic natural mutational frequencies. Starting with the BA.1 sequence, mutational frequencies from 2021 and 2022 were used to generate ten sets of 250,000 synthetic RBD sequences through six rounds of in silico evolution, where the 100 variants with the highest predicted score for ACE2 binding (averaged across the ensemble CNN models) in each round were used as seed sequences for the next round of mutations. Next, the ensemble deep learning models were used to predict antibody binding or escape for the synthetic RBD variants. This provides an estimation of each individual antibody's binding breadth in the generated sequence space and thus correlates with resistance to prospective Omicron lineages (Fig. 5a,b and Supplementary Fig. 9).

As several of the clinically used antibody therapies for COVID-19 consisted of a cocktail of two antibodies (such as LY-CoV555 + LY-CoV16, REGN10933 + REGN10987 and COV2-2130 + COV2-2196), we also determined antibody breadth across all two-way combinations. For the 2022-based synthetic lineages, ZCB11 showed the greatest predicted breadth, followed by A23-58.1, Brii-198 and ADG20 (Fig. 5b). The predicted coverage of ZCB11 corresponds well with experimental measurements that show it maintains high affinities and neutralization to several Omicron variants (BA.2, BA.4/5)[10]. Similarly, Brii-198 and A23-58.1 have been shown to bind BA.2, BA.2.12 and BA.2.75 variants[40], aligning with the predictions of their relatively high breadth. Examining breadth profiles of each antibody as a function of ED revealed differing profiles, such as ZCB11 and Brii-198 maintaining high breadth at larger ED ($>ED_4$), while A23-58.1 and ADG20 have substantially lower breadth at large ED (Fig. 5c). The predicted breadth of several antibodies were substantially different for synthetic lineages generated using 2021 mutational probabilities. For example, the breadth of ADG20 is substantially higher as it is predicted to bind 16% more variants, while the breadth of Brii-198 and A23-58.1 are both reduced by 11% (Supplementary Fig. 10). This suggests that correctly anticipating antigenic drift and changes in mutational frequencies play an important role in determining breadth predictions.

It is worth noting that calculating the breadth of antibody combinations is not simply additive. For example, while Brii-198 ranks lower than A23-58.1 in total breadth, Brii-198 provides more complementary coverage to ZCB11 (Brii-198 binds to more variants that escape ZCB11), resulting in an overall increase in variant coverage in a simulated cocktail. Thus, when designing a cocktail, to select the best complementary antibody, both the quantity and additional qualities (such as mutational patterns) of escape variant lineages that are 're-captured' by an antibody must be considered (Fig. 5d). Examining the distribution of escape variants for ZCB11 at $ED_6$, where it sees its most substantial breadth reduction, the three other highly ranked antibodies (A23-58.1, Brii-198 and ADG20) re-establish coverage (predicted binding) over unique lineages, with Brii-198 re-capturing the greatest number of high-distance escape variants to ZCB11 (Fig. 5d,e). Taking a closer look at the mutations within the re-captured sequences, only ADG20 and Brii-198 cover and mitigate variants that include the key F486V mutation (for example, BA.4/5). Furthermore, Brii-198 covers the most

**Fig. 5 | Evaluating antibody breadth on synthetic Omicron lineages.**
**a**, Example of a synthetic lineage tree of sequences generated containing mutations unseen in major Omicron variants, with heat map indicating the deep learning predictions of binding or escape for individual antibodies. VOCs, variants of concern. **b**, Total mean predicted breadth of individual antibodies and combinations on synthetic lineages generated from 2022 mutational probabilities. **c**, The fraction (%) of sequences bound by individual antibodies at different ED from BA.1. **d**, Phylogenetic tree of ZCB11 escape

variants containing the 20 highest-scoring mutations, with antibody recapture indicated. **e**, Sequence logos show the mutations in the top 25 positions with greatest KL divergence in ZCB11 escape variants at $ED_6$, and number of sequences re-captured by Brii-198, ADG20 and A23-58.1. (Higher recapture indicates a more complementary antibody). **f**, The top 50 predicted mutations ranked by their escape scores (Methods) from the generated synthetic lineages, with new mutations seen in the BA.2.86 variant highlighted.

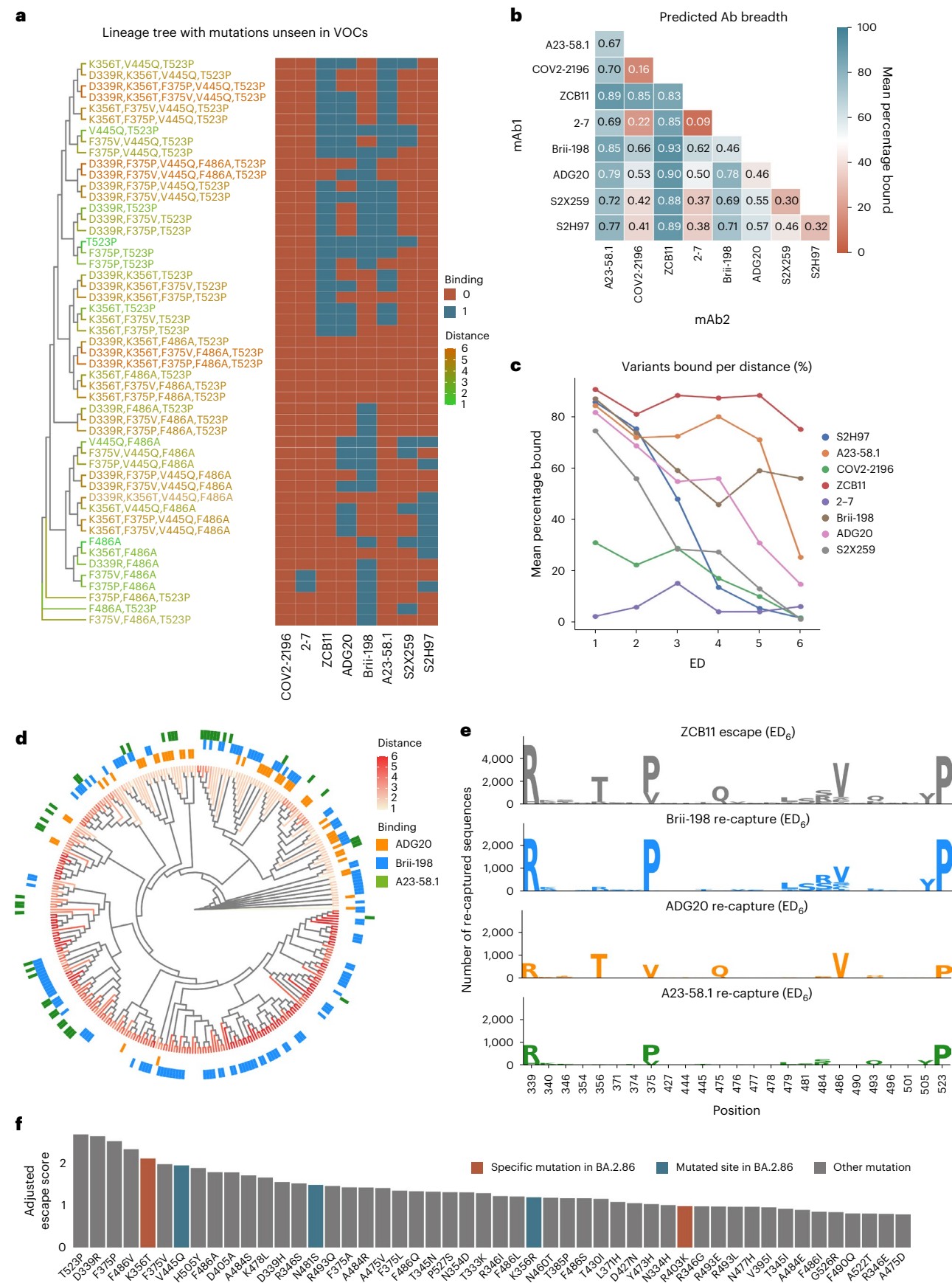

diverse sequences that contain additional critical mutations at the F468 position, in addition to the surrounding residues in this epitope (Fig. 5e). Thus, while any of the three antibodies would be complementary to ZCB11 by nature of targeting a different epitope[10], our breadth analysis aids in identifying the most complementary antibody based on RBD variant coverage.

To quantify the impact of how individual mutations can drive antibody escape, an escape score ($S\_m\hat{}$) was computed for each mutation (*m*) within the synthetic lineages. This metric is a normalized product of the number of antibodies escaped by a given mutation and the mutation's frequency within the lineage (Methods). When examining individual RBD mutations across the synthetic lineages (Fig. 5f), it was revealed that T523P has the highest escape score. Comparatively, DMS results showed that mutations at position 523 have a slightly negative influence on RBD protein expression level[19], which may explain its low occurrence in natural variants, having only been observed in 70 sequences in the GISAID database. Furthermore, the combination of D339R, F486A and T523P mutations in the simulated BA.1 lineages caused the most antibody escape among mutations not previously observed in major variants (Fig. 5f). Out of these, the positions 339 and 486 are mutated in BA.2.75 and XBB and their sublineages. The top 50 mutations with the highest escape scores include K356T and R403K, which are present in the recently reported and highly mutated BA.2.86 variant and had not been previously reported in any other major variant (Fig. 5f). In addition, positions V445 and N481 were also mutated in BA.2.86. Taken together, this suggests that DML-derived escape scores may reveal mutations or positions that emerge in future variants.

## Discussion

The emergence of SARS-CoV-2 lineages with a high number of mutations has resulted in substantial viral immune evasion, including ineffective neutralization by previously developed therapeutic antibodies[5]. This rapid pace of viral evolution has underscored the need for novel approaches to adequately profile antibody candidates and predict their robustness to emerging variants early on during drug development. To this end, we leverage DML, a machine-learning-guided protein engineering method to prospectively evaluate clinically relevant antibodies for their breadth against potential future Omicron variants across a large mutational sequence space.

We first demonstrate the feasibility of assembling full-length RBD mutagenesis libraries with high fidelity using a large number of relatively short ssODNs in a one-pot reaction and obtaining library sizes in excess of $10^8$. This is despite the fact that previous studies have reported a decrease in GGA efficiency when increasing the number of DNA fragments[54]. Screening of these libraries for ACE2 binding and antibody escape yielded high-dimensional data sets with combinatorial mutations spanning the entire RBD sequence, which is not obtainable through frequently used approaches such as DMS. In addition, the RBD library design can be updated to accommodate mutations present in emerging variants, and the average number of mutations can be titrated to generate data suitable for the training of machine learning models. This library design and screening approach could also be exploited to profile viral surface proteins from other rapidly evolving viruses such as influenza or HIV, two viruses which undergo substantial antigenic drift that drives their immune escape[55–57].

So far, the breadth of SARS-CoV-2 therapeutics has been assessed through the use of past variants and observed mutations[20,58–60]. Measuring breadth in this way does not adequately predict long-term resistance against future variants. The deployment of ensemble deep learning models to make predictions on synthetic mutational trajectories of the RBD enabled an effective quantitative method to evaluate the breadth of each antibody based on its coverage of RBD mutational sequence space. DML predictions confirm that ZCB11 has exceptionally broad breadth to major Omicron lineages that emerged in 2022, while many other antibodies fail against Omicron variants[39]. Furthermore,

our results suggest that the standard structure-based approach of selecting antibodies targeting different epitopes in a cocktail does not sufficiently determine which combinations offer the most cumulative breadth. High breadth cocktails would ensure that even if a variant escapes one antibody in the cocktail, it has a high chance to be re-captured by the other antibody—thus potentially maintaining the clinical effectiveness of the therapy. For example, this occurred with the combination antibody therapy from Eli Lilly (LY-CoV555 + LY-CoV16), which continued to be used clinically when only a single antibody in the combination was effective after the emergence of Beta, Gamma and Delta variants[22,61]. It is worth noting that a comprehensive search through a SARS-CoV-2 antibody database (Cov-AbDab, accessed April 2023)[62] reveals that a number of neutralizing antibodies discovered early in the pandemic from patients infected with the ancestral Wu-Hu-1 are still able to neutralize Omicron variants such as BA.5, BQ.1 and XBB.1. DML could therefore be a powerful tool to identify such variant-resistant antibodies for therapeutic development.

Analysis of DML breadth predictions also highlights specific and positional mutations that are associated with greater immune escape, with four such mutations being observed in the recently discovered and highly mutated BA.2.86 variant. By contrast, other recently published deep learning methods, which rely on models trained using a combination of DMS and protein structure data, were able to only correctly forecast one new mutation each that appeared in the XBB.1.5 and BQ.1 variants, respectively[30,31,63]. While this shows the value of using protein structural information to better infer higher-order effects between mutations, these models are still limited by the use of low-distance (most often single-mutation) DMS data. Thus, it would be worthwhile to explore whether the use of combinatorial DML data can further improve the accuracy and forecasting performance of models trained using a multi-task objective, similar to those mentioned above.

A current limitation that faces the DML approach is a sensitivity issue when it comes to making predictions for low-affinity antibodies. Future work can explore the possibility of sorting at multiple antibody concentrations and building multi-label or regression models to predict quantitative changes in antibody affinity to given variants, rather than a binding/non-binding label obtained from our current classifiers. The resulting predictions would be more nuanced in cases where antibody affinities are already weak, such as antibodies 2–7 and COV2-2196.

Finally, the accuracy of antibody breadth predictions is dependent on having an accurate forecast of future mutations in the RBD. The use of deep learning models that predict ACE2 binding allowed us to capture evolutionary pressures correlated with host receptor binding, which is a mandatory feature of any emerging SARS-CoV-2 variant[64]. However, a myriad of other factors impact antigenic drift and variant emergence, such as transmissibility, host cell infectivity, crossover and reproductive rate[65], thus generating training data related to these factors, for example, through the use of an advanced pseudovirus mutational library screening system[66], may further support the generation of deep learning models that can predict future mutations and variants with higher accuracy.

## Methods

### Construction of a high-distance Omicron RBD library for yeast surface display

Synthetic ssODNs (oligo pool from IDT) were designed with either one or all possible combinations of two degenerate NNK codons for each fragment (Supplementary Table 1). For each fragment, 137 ssODNs were designed (1 wild type, 16 with single NNK codons and 120 double NNK codon combinations). Each fragment was flanked by BsmBI recognition sites and ~20 nucleotides for second-strand synthesis through PCR. For high-fidelity library assembly, the overhangs were optimized using the New England Biolabs ligase fidelity viewer (https://ligasefidelity.neb.com/viewset/run.cgi). Using the NEBridge Golden Gate Assembly Kit (NEB, E1602), individual fragments were assembled to full-length RBD

gene segments. A custom entry vector based on pYTK001 (addgene, Kit 1000000061) was designed. Double-stranded fragments were mixed with 75 ng entry vector in a 2:1 molar ratio. As suggested by the manufacturer's instructions, 2 μl NEB Golden Gate Enzyme Mix was used. For the assembly, the following protocol was used: (42 °C, 5 min to 16 °C, 5 min) × 30, to 60 °C, 5 min. The assembled libraries were transformed into *Escherichia coli* DH5α ElecroMAX (Thermo Fisher Scientific, 11319019), resulting in ~4 × 10$^8$ transformants. According to the manufacturer's instructions (Zymo, D4201), the RBD library plasmid was extracted from *E. coli*.

The RBD library was PCR amplified, and the yeast display vector (pYD1) was linearized using the restriction enzyme BamHI (Thermo Fisher Scientific, FD0054). Both insert and backbone were column purified according to the manufacturer's instructions (D4033) and drop dialysed for 2 h using nuclease-free water (Millipore VSWP02500). The RBD library insert and linearized pYD1 backbone were co-transformed into yeast (*S. cerevisiae*, strain EBY100) using a previously described protocol[67]. Briefly, EBY100 (ATCC, MYA-4941) was grown overnight in YPD (20 g l$^{-1}$ glucose (Sigma-Aldrich, G8270), 20 g l$^{-1}$ vegetable peptone (Sigma-Aldrich, 19942) and 10 g l$^{-1}$ yeast extract (Sigma-Aldrich, Y1625) in deionized water). On the day of the library preparation, yeast cells from the overnight culture were inoculated in 300 ml YPD at an optical density at 600 nm of 0.3. The cells were grown to an optical density at 600 nm of 1.6 before washing the cells twice with 300 ml ice-cold 1 M sorbitol solution (Sigma-Aldrich, S1876). In a subsequent step, the cells were conditioned using a solution containing 100 mM lithium acetate (Sigma-Aldrich, L6883) and 10 mM DTT (Roche, 10197777001) for 30 min at 30 °C. This was followed by a third wash using 300 ml ice-cold 1 M sorbitol solution. Using 50 μg insert and 10 μg pYD1 backbone, electrocompetent EBY100 was transformed using 2 mm electroporation cuvettes (Sigma-Aldrich, Z706086). The cells were recovered for 1 h in recovery medium (YPD: 1 M sorbitol solution mixed in a 1:1 ratio) before passaging the cells into selective SD-CAA medium (20 g l$^{-1}$ glucose (Sigma-Aldrich, G8270), 8.56 g l$^{-1}$ NaH$_2$PO$_4$·H$_2$O (Roth, K300.1), 6.77 g l$^{-1}$ Na$_2$HPO$_4$·2H$_2$O (Sigma-Aldrich, 1.06580), 6.7 g l$^{-1}$ yeast nitrogen base without amino acids (Sigma-Aldrich, Y0626) and 5 g l$^{-1}$ casamino acids (Gibco, 223120) in deionized water). The cells were grown for 2 days at 30 °C. To estimate the transformation efficiency, dilution plating was performed. Approximately 2 × 10$^8$ transformants were obtained.

### Screening RBD libraries for ACE2-binding or non-binding
Yeast cells containing the RBD library plasmid were grown in SD-CAA for 18–24 h at 30 °C. Surface display of Omicron RBD was induced by passaging the cells into SG-CAA medium (20 g l$^{-1}$ galactose (Sigma-Aldrich, G0625), 8.56 g l$^{-1}$ NaH$_2$PO$_4$·H$_2$O, 6.77 g l$^{-1}$ Na$_2$HPO$_4$·2H$_2$O, 6.7 g l$^{-1}$ yeast nitrogen base without amino acids and 5 g l$^{-1}$ casamino acids in deionized water). The cells were incubated at 23 °C for 48 h, as previously described[37]. Approximately 10$^9$ cells were spun down by centrifugation at 3,500 × g for 3 min and washed once with 5 ml cold wash buffer (DPBS (PAN Biotech, P04-53500) + 0.5% BSA (Sigma-Aldrich, A2153) + 2 mM EDTA (Biosolve, 051423) + 0.1% Tween20 (Sigma Aldrich, P1379)). Next, cells were labelled with 50 nM of biotinylated human ACE2 protein (Acro Biosystems, AC2-H82E6) for 30 min at 4 °C at 700 r.p.m. on a shaker (Eppendorf, ThermoMixer C). About 50 nM ACE2 has been proven in other studies to be an adequate concentration[32]. As the reported affinities for various natural RBD variants for ACE2 are in the single- to double-digit nanomolar range, for example, Wuhan (27.5 nM), Alpha (6.7 nM), Beta (19.7 nM), Gamma (16.0 nM), Delta (25.1 nM) or Omicron (5.5 nM)[68,69], it is expected that the affinities of the yeast-display RBD variants that can be detected as binders at 50 nM ACE2 are in a physiological range consistent with infectious SARS-CoV-2 variants high enough to be considered infectious. The cells were subsequently washed. In a secondary staining step, cells were labelled with streptavidin–phycoerythrin (PE) (Biolegend 405203) (1:80 diluted) and anti-FLAG Tag Allophycocyanin (APC) (Biolegend, 637308) (1:200

dilution) at 4 °C for 30 min at 700 r.p.m. Afterwards, cells were centrifuged at 3,500 × g for 3 min. The supernatant was discarded, and the tube was protected from light and stored on ice until sorting. Binding (PE+/APC+) and non-binding (PE−/APC+) populations of yeast cells were collected by FACS (BD FACSAria Fusion or BD Influx) (Fig. 3a,b and Supplementary Fig. 2). Shown FACS plots depict 10$^4$ events. Figure 3a shows a representative example of the sorting strategy. Collected cells were pelleted at 3,500 × g for 3 min to remove the FACS buffer. The cells were resuspended using SD-CAA and grown for 2 days at 30 °C. The sorting process was repeated until the desired populations were pure.

### Screening RBD libraries for antibody binding or escape
The ACE2-binding population of yeast cells expressing the RBD library was grown and induced as described above. Approximately 10$^8$ cells were pelleted by centrifugation at 3,500 × g for 3 min at 4 °C and washed once with 1 ml wash buffer. The washed cells were incubated with antibodies (concentrations listed in Supplementary Table 2). Suitable concentrations approximately corresponding to the EC$_{90}$ were experimentally determined beforehand (Supplementary Fig. 3). Cells were incubated for 30 min at 4 °C and 700 r.p.m. After an additional washing step, a secondary stain was performed using 5 ng ml$^{-1}$ anti-human IgG-AlexaFluor647 (AF647) (Jackson Immunoresearch, 109-605-098) (1:200 dilution). The cells were incubated for 30 min at 4 °C and 700 r.p.m. Subsequently, cells were washed and stained in a tertiary staining step using 1 ng ml$^{-1}$ anti-FLAG-PE (1:200 dilution) for 30 min at 4 °C and 700 r.p.m. Cells were pelleted by centrifugation at 3,500 × g for 3 min at 4 °C. The supernatant was discarded, and the tube was protected from light and stored on ice until sorting. Cells expressing RBD that maintained antibody-binding (AF647+/PE+) or showed a complete loss of antibody binding (AF647−/PE+) were isolated using FACS (BD Aria Fusion or Influx BD). Shown FACS plots depict 10$^4$ events. Figure 3b shows a representative example of the sorting strategy. Collected cells were pelleted by centrifugation at 3,500 × g for 3 min at room temperature. The FACS buffer was discarded, and the cells were resuspended using SD-CAA. The cells were cultured for 48 h at 30 °C. The sorting process was repeated once for the binding population and twice for the non-binding population. This procedure yielded pure binding and non-binding (escape) populations.

### Deep sequencing of RBD libraries
The pYD1 plasmid encoding the RBD library was extracted from yeast cells per manufacturer's instructions (Zymo, D2004). The mutagenized part of the RBD was PCR amplified using custom-designed primers for seq-library A and seq-library B (Supplementary Table 6). In a second PCR amplification step, sample-specific barcodes (Illumina Nextera) were introduced, which allowed pooling of individual populations for sequencing. The populations were sequenced using the Illumina MiSeq v 3 kit which allows for 2 × 300 paired-end sequencing.

### Preprocessing of deep sequencing data
Sequencing reads were paired, quality trimmed and merged using the BBTools suite[70] with a quality threshold of qphred $R$ > 25. RBD nucleotide sequences were then extracted using custom R scripts, followed by translation to amino acid sequences. Read counts per sequence were calculated, and singletons (read count = 1) were discarded. Sequencing datasets used for training machine and deep learning models were created by combining the binding and non-binding datasets. Sequences present in both populations were removed.

Binding scores for heat maps shown in Fig. 3c–e were created by calculating amino acid counts per position in the RBD from both binding and non-binding sequences. Wild-type (BA.1) amino acid residues were then removed, relative frequencies were calculated with a pseudocount of 1 added, and final binding scores were calculated as binding frequencies divided by non-binding frequencies. The results were then log-transformed before plotting in the heat map for visualization.

## Training and testing machine and deep learning models

All machine learning code and models were built in Python (3.10.4)[71]. For data processing and visualization, numpy (1.23.3), pandas (1.4.4), matplotlib (3.5.3) and seaborn (0.12.0) packages were used. Baseline benchmarking models were built using Scikit-Learn (1.0.2), while Keras (2.9.0) and Tensorflow (2.9.1) were used to build the MLP and CNN models.

Each model was trained using 80/10/10 train–val–test data random splits. RBD library protein sequences (from seq-library A or B deep sequencing data) were one-hot encoded before being used as inputs into the models. For the CNN, the 2D one-hot encoded matrix was used as the input, while for others, the matrix was flattened into a one-dimensional vector. All reported model performances were evaluated using fivefold cross-validation and evaluated based on the metrics for accuracy, F1, MCC, precision and recall.

When training baseline machine learning models, class balancing was performed on the training set through random upsampling of the minority class, while validation and test sets were kept unbalanced. Hyperparameter optimization was performed during model training using up to 30 rounds of RandomSearchCV (from Scikit-Learn), and the performance of the best models were used for final comparisons.

To train the final deep learning models, exhaustive hyperparameter search was performed on the CNN models to optimize performance through the hyperparameters listed in Supplementary Table 4. The training dataset was balanced at different ratios (Minority Ratio row in Supplementary Table 4), while validation and test sets remained unbalanced to appropriately evaluate MCC, precision and recall scores on imbalanced data. Dataset balancing was performed through rejection sampling using a custom dataset sampler created in Tensorflow. To prevent data leakage during training of the models for ensembles, the held-out test set was fixed, while multiple models were trained on random splits of the training and validation sets to make sure each model learned slightly different parameters of the dataset, while being evaluated on the same held-out test sequences.

## Predictions made with ensemble deep learning models

Natural and in silico generated synthetic RBD variant sequences were assigned binding, escape and 'uncertain' labels for ACE2 and antibodies using an ensemble of trained models. For a given RBD sequence, each model assigns a binding label if output $P > 0.5$, escape if output $P \leq 0.5$. For each of the two libraries (seq-libraries A and B), the three models with the highest MCC scores were used to independently assign labels to each sequence, followed by majority voting, where the most common label was taken as the label for each variant. The labels from models trained with seq-library A or seq-library B were used to determine the final label for each variant: binding if both libraries agree on a binding label and escape if either library predicts escape. For experimentally measured variants collected from publications, antibody-variant pairs were labelled as escape if their measured $K_d$ was >100 nM or $IC_{50}$ was >10 µg ml⁻¹ (Supplementary Data 2).

## Calculating mutational probabilities of the RBD based on SARS-CoV-2 genome data

To generate the mutational probability matrices used for synthetic lineages, SARS-CoV-2 spike protein sequences were obtained from the GISAID database (most recent access of June 2023). The regions corresponding to the RBD were extracted, along with the date when each sequence was deposited into the database. Sequences were separated by the year they were added (for example, 2021 or 2022). From these sequences, mutations were counted at each position per position and per amino acid. Mutational frequencies at each position were calculated using these counts. Finally, a log softmax function was applied to obtain mutational probabilities for each position. For each position, only residues that were observed in GISAID sequences were counted, while all unseen residues were not included in the softmax transform, preventing them from being generated in synthetic lineages.

## In silico generation of synthetic Omicron lineages

Using BA.1 as the initial seed variant, in silico sequences were generated in a stepwise fashion over six rounds of mutations. In the first round, single mutations were randomly generated across the RBD. Positions and amino acid for each mutational round were selected using probabilities from the 2021 or 2022 substitution matrices; as a control, sequences were also generated using no substitution matrix (where all mutations were sampled from a uniform probabilities distribution). Then binding probability scores were assigned to variants in each generation by taking the average of all $P$ predicted by each of the ACE2 models in the ensemble. The top 100 variants ranked by ACE2-binding $P$ were used as seed sequences for the next round of mutations. For each round, new variants were only accepted if they contained mutations not previously seen in other generated variants, or else the process was repeated again and new mutations selected until the maximum number of variants were reached (250,000).

## Calculating escape scores

An escape score ($S_m$) was calculated that aims to quantify the impact of a given mutation on driving escape from the antibodies tested herein and was calculated using the following:

$$S_m = \frac{\sum_{E=0}^{a} \frac{(E \times f \times \underline{d})}{n}}{N}$$

$S_m$ is the escape score of a mutation $m$, $E$ is the number of antibodies that are predicted to escape from $m$, and within the group of sequences with the same number of $E$, $f$ is the frequency that $m$ appeared in the sequence group, $\underline{d}$ is the mean of sequence ED from BA.1, $n$ is the number of sequences, $N$ is how many times one mutation appeared in different groups of $E$, and $a$ is the total number of groups, according to how many antibodies were tested (here, $a = 8$). For better visualization, the adjusted escape score was used (Fig. 5a,f) and is calculated with the following equation:

$$S_{adj} = \log_{10} S_m + 7.$$

## Additional statistical analysis and plots

Statistical analysis was performed using Python (3.10.4) with the Scipy package (1.9.3). Dimensionality reduction was performed using UMAP-learn (0.5.3). Graphics were generated using matplotlib (3.5.3), seaborn (0.12.0) and ggtree (3.8.0). Sequence logo plots were created using Seq2Logo (5.29.8)[72] or the dsmlogo package from the Bloom Lab (https://github.com/jbloomlab/dmslogo).

The Kullback–Leibler (KL) divergence was calculated by adapting a recently described method[30]. In short, a probability-weighted KL logo plot was used to visualize differences between a subset of sequences to the full background dataset. Let $M_1 = (f_1, f_2, f_3, ..., f_n)$ represent the position frequency matrix of the background sequence set, where the length of the initial sequence is $n = 201$ and each frequency $f_i = (a_1, a_2, a_3, ... a_{20})^T$ represents the frequency of each amino acid per position $i$. At the same time, $M_2 = (f_1', f_2', f_3', ... f_n')$ represents the position frequency matrix of the subset of sequences, each $f_i' = (a_1', a_2', a_3', ... a_{20}')^T$. The KL divergence at each position is computed as follows:

$$D_{KL}(f_i' \parallel f_i) = \sum_{i=1}^{20} a_i' \cdot \ln \ln \left( \frac{a_i'}{a_i} \right)$$

The height and direction of each amino acid letter are calculated through probability-weighted normalization as part of the Seq2Logo package using the following:

$$h(a_i') = \frac{a_i' \left( \frac{a_i'}{a_i} \right)}{\sum_{i=1}^{20} a_i' \cdot \left| \ln \ln \left( \frac{a_i'}{a_i} \right) \right|} D_{KL}(f_i' \parallel f_i)$$

## Reporting summary

Further information on research design is available in the Nature Portfolio Reporting Summary linked to this article.

## Data availability

The main data supporting the results in this study are available within the paper and its Supplementary Information. The processed datasets generated during the study are available via Zenodo at https://doi.org/10.1101/2023.10.09.561492 (ref. 73). Source data are provided with this paper.

## Code availability

The code and models used to perform the work in this study are available at https://github.com/LSSI-ETH/Omicron_DML.

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

## Acknowledgements

We thank the ETH Zurich Department of Biosystems Science and Engineering Single Cell Unit and the ETH Zurich Department of Biosystems Science and Engineering Genomics Facility for support. This work was also supported by the Basel Research Centre for Child Health (FTC COVID-19, to S.T.R.).

## Author contributions

L.F., B.G., J.H., J.M.T. and S.T.R. developed the methodology. L.F. and J.M.T. designed and generated mutagenesis libraries. L.F. performed screening experiments and sequencing. B.G. and J.H. analysed the sequencing data and performed deep-learning analyses. L.F., B.G., J.H. and S.T.R. wrote the manuscript, with input from all other authors.

## Funding

## Competing interests

C.R.W. is an employee of Alloy Therapeutics (Switzerland). C.R.W. and S.T.R. hold shares of Alloy Therapeutics. S.T.R. is on the scientific advisory board of Alloy Therapeutics. The other authors declare no competing interests.

## Additional information

**Correspondence and requests for materials** should be addressed to Sai T. Reddy.

# Reporting Summary

## Statistics

For all statistical analyses, confirm that the following items are present in the figure legend, table legend, main text, or Methods section.

| n/a | Confirmed | |
|---|---|---|
| ☐ | ☒ | The exact sample size (*n*) for each experimental group/condition, given as a discrete number and unit of measurement |
| ☐ | ☒ | A statement on whether measurements were taken from distinct samples or whether the same sample was measured repeatedly |
| ☒ | ☐ | The statistical test(s) used AND whether they are one- or two-sided<br>*Only common tests should be described solely by name; describe more complex techniques in the Methods section.* |
| ☒ | ☐ | A description of all covariates tested |
| ☒ | ☐ | A description of any assumptions or corrections, such as tests of normality and adjustment for multiple comparisons |
| ☒ | ☐ | A full description of the statistical parameters including central tendency (e.g. means) or other basic estimates (e.g. regression coefficient) AND variation (e.g. standard deviation) or associated estimates of uncertainty (e.g. confidence intervals) |
| ☒ | ☐ | For null hypothesis testing, the test statistic (e.g. *F*, *t*, *r*) with confidence intervals, effect sizes, degrees of freedom and *P* value noted<br>*Give P values as exact values whenever suitable.* |
| ☒ | ☐ | For Bayesian analysis, information on the choice of priors and Markov chain Monte Carlo settings |
| ☒ | ☐ | For hierarchical and complex designs, identification of the appropriate level for tests and full reporting of outcomes |
| ☒ | ☐ | Estimates of effect sizes (e.g. Cohen's *d*, Pearson's *r*), indicating how they were calculated |

*Our web collection on statistics for biologists contains articles on many of the points above.*

## Software and code

Policy information about availability of computer code

| | |
|---|---|
| Data collection | Illumina software and FASTQC were used for the initial processing and quality assessment of NGS datasets. |
| Data analysis | Custom code written in Python and R have been used for data analysis. The code associated with the work in this study is available at https://github.com/LSSI-ETH/Omicron_DML. |

For manuscripts utilizing custom algorithms or software that are central to the research but not yet described in published literature, software must be made available to editors and reviewers. We strongly encourage code deposition in a community repository (e.g. GitHub). See the Nature Portfolio guidelines for submitting code & software for further information.

## Data

Policy information about availability of data

All manuscripts must include a data availability statement. This statement should provide the following information, where applicable:
- Accession codes, unique identifiers, or web links for publicly available datasets
- A description of any restrictions on data availability
- For clinical datasets or third party data, please ensure that the statement adheres to our policy

The main data supporting the results in this study are available within the paper and its Supplementary Information. Source data are provided with this paper. The processed datasets generated during the study are available at https://zenodo.org/records/11172179.

## Research involving human participants, their data, or biological material

Policy information about studies with human participants or human data. See also policy information about sex, gender (identity/presentation), and sexual orientation and race, ethnicity and racism.

| | |
|---|---|
| Reporting on sex and gender | The study did not involve human research participants. |
| Reporting on race, ethnicity, or other socially relevant groupings | – |
| Population characteristics | – |
| Recruitment | – |
| Ethics oversight | – |

Note that full information on the approval of the study protocol must also be provided in the manuscript.

# Field-specific reporting

Please select the one below that is the best fit for your research. If you are not sure, read the appropriate sections before making your selection.

☒ Life sciences ☐ Behavioural & social sciences ☐ Ecological, evolutionary & environmental sciences

For a reference copy of the document with all sections, see nature.com/documents/nr-reporting-summary-flat.pdf

# Life sciences study design

All studies must disclose on these points even when the disclosure is negative.

| | |
|---|---|
| Sample size | 8 different monoclonal antibodies and human ACE2 were investigated for binding/non-binding to a library of Omicron BA.1 variants. E. coli and yeast library sizes were determined by dilution plating. |
| Data exclusions | No data were excluded. |
| Replication | The methods and materials used for data generation, collection and analysis are described in detail in Methods and were reproducible. |
| Randomization | Sequences selected during the training and assessment of models were randomly subsampled from the full experimental datasets. No other randomization was relevant or required for the remaining experiments performed. |
| Blinding | Blinding was not relevant. |

# Reporting for specific materials, systems and methods

We require information from authors about some types of materials, experimental systems and methods used in many studies. Here, indicate whether each material, system or method listed is relevant to your study. If you are not sure if a list item applies to your research, read the appropriate section before selecting a response.

### Materials & experimental systems

| n/a | Involved in the study |
|---|---|
| ☐ | ☒ Antibodies |
| ☐ | ☒ Eukaryotic cell lines |
| ☒ | ☐ Palaeontology and archaeology |
| ☒ | ☐ Animals and other organisms |
| ☒ | ☐ Clinical data |
| ☒ | ☐ Dual use research of concern |
| ☒ | ☐ Plants |

### Methods

| n/a | Involved in the study |
|---|---|
| ☒ | ☐ ChIP-seq |
| ☐ | ☒ Flow cytometry |
| ☒ | ☐ MRI-based neuroimaging |

## Antibodies

| | |
|---|---|
| Antibodies used | A23-58.1, COV2-2196, Brii-198, ZCB11, 2-7, S2X259, ADG20, S2H97 anti-human IgG-AF647 (Jackson Immunoresearch 109-605-098), anti-FLAG Tag APC (Biolegend 637308). |

| Validation | Antibodies produced in-house were validated by testing for binding to the Omicron BA.1 sequence.<br>Antibodies purchased from BioLegend were quality-control tested by intracellular immunofluorescent staining with flow cytometric analysis. |
|---|---|

## Eukaryotic cell lines

Policy information about cell lines and Sex and Gender in Research

| Cell line source(s) | EBY100 (ATCC). |
|---|---|
| Authentication | The EBY100 yeast cell line was authenticated via the successful cell-surface display of RBD. |
| Mycoplasma contamination | The cells tested negative for mycoplasma contamination. |
| Commonly misidentified lines<br>(See ICLAC register) | No commonly misidentified cell lines were used. |

## Flow Cytometry

### Plots

Confirm that:

☒ The axis labels state the marker and fluorochrome used (e.g. CD4-FITC).

☒ The axis scales are clearly visible. Include numbers along axes only for bottom left plot of group (a 'group' is an analysis of identical markers).

☒ All plots are contour plots with outliers or pseudocolor plots.

☒ A numerical value for number of cells or percentage (with statistics) is provided.

### Methodology

| Sample preparation | The Omicron RBD library was displayed using the yeast strain EBY100. Cells were grown for 1 day in selective medium at 30 °C before inducing surface display for 2 days at 23 °C. After completion of induction, the cells were washed before incubation with ACE2/monoclonal antibody for 1 h at 4 °C. The cells were washed before incubation with anti-FLAG-PE and anti-IgG-AF647. The cells were washed once more before sorting. |
|---|---|
| Instrument | Flow cytometry was performed using BD FACSAria Fusion (5 lasers configurations, UV, 488, 405, 445, 561, 640) and BD Influx (8 lasers configurations, 488nm, 405nm, 445nm, 514nm, 532nm, 561nm, 594nm and 640nm). |
| Software | The data were analysed using FlowJo v10.9. |
| Cell population abundance | The post-sort purity of populations was determined by flow-cytometry analysis following expansion of the collected populations. 2 or 3 rounds of sorting were necessary to isolate pure populations. |
| Gating strategy | The FSC/SSC gate showed a single population. An FSC-H/FSC-W gate was used to distinguish singlets from doublets. Cell doublets were excluded. RBD expressing cells were then identified by their fluorescence signal from the 561 nm laser (586/15 nm band-pass filter, 570 nm longpass filter). Cells expressing antibody binding RBD variants were then identified and gated determined by their fluorescence signal from 633 nm laser (670/14 nm band-pass filter). Boundaries between positive and negative populations were determined by including the following control samples in the analysis: 1) a yeast line expressing wild-type Omicron BA.1 RBD stained with either ACE2 or monoclonal antibody. 2) a yeast line expressing wild-type Omicron BA.1 RBD stained only for expression and not with ACE2/monoclonal antibody.<br><br>The population with intermediate binding signal and the double negative population were not collected. |

☒ Tick this box to confirm that a figure exemplifying the gating strategy is provided in the Supplementary Information.

