## [Peer Review File · Nature Biomedical Engineering]

Deep mutational learning for the selection of therapeutic antibodies resistant to the evolution of Omicron variants of SARS-CoV-2

Corresponding author: Sai Reddy

Editorial note

This document includes relevant written communications between the manuscript's corresponding author and the editor and reviewers of the manuscript during peer review. It includes decision letters relaying any editorial points and peer-review reports, and the authors' replies to these (under 'Rebuttal' headings). The editorial decisions are signed by the manuscript's handling editor, yet the editorial team and ultimately the journal's Chief Editor share responsibility for all decisions.

Any relevant documents attached to the decision letters are referred to as **Appendix #**, and can be found appended to this document. Any information deemed confidential has been redacted or removed. Earlier versions of the manuscript are not published, yet the originally submitted version may be available as a preprint. Because of editorial edits and changes during peer review, the published title of the paper and the title mentioned in below correspondence may differ.

Correspondence

Thu 04 Jan 2024

Decision on Article nBME-23-2751

Dear Prof Reddy,

Thank you again for submitting to *Nature Biomedical Engineering* your manuscript, "Deep learning-guided selection of antibody therapies with enhanced resistance to current and prospective SARS-CoV-2 Omicron variants", and for your patience in waiting for the reviewer feedback.

The manuscript has been seen by three experts, whose reports you will find at the end of this message. You will see that the reviewers appreciate the work, and that they provide useful suggestions and raise pertinent questions. I am hoping that a revised version of the manuscript will appropriately respond to the queries and satisfy the reviewers. Also, please ensure that all raw and analysed datasets are deposited, to facilitate re-use.

When you are ready to resubmit your manuscript, please upload the revised files, a point-by-point rebuttal to the comments from all reviewers, the reporting summary, and a cover letter that explains the main improvements included in the revision and responds to any points highlighted in this decision.

Please follow the following recommendations:

- * Clearly highlight any amendments to the text and figures to help the reviewers and editors find and understand the changes (yet keep in mind that excessive marking can hinder readability).
- * If you and your co-authors disagree with a criticism, provide the arguments to the reviewer (optionally, indicate the relevant points in the cover letter).
- * If a criticism or suggestion is not addressed, please indicate so in the rebuttal to the reviewer commentsand explain the reason(s).

* Consider including responses to any criticisms raised by more than one reviewer at the beginning of the rebuttal, in a section addressed to all reviewers.

* The rebuttal should include the reviewer comments in point-by-point format (please note that we provide all reviewers will the reports as they appear at the end of this message).

* Provide the rebuttal to the reviewer comments and the cover letter as separate files.

We hope that you will be able to resubmit the manuscript within 8 weeks from the receipt of this message. If this is the case, you will be protected against potential scooping. Otherwise, we will be happy to consider a revised manuscript as long as the significance of the work is not compromised by work published elsewhere or accepted for publication at *Nature Biomedical Engineering*.

We hope that you will find the referee reports helpful when revising the work, which we look forward to receive. Please do not hesitate to contact me should you have any questions.

Best wishes,

Pep

Pep Pàmies
Chief Editor, Nature Biomedical Engineering

Reviewer #1 (Report for the authors (Required)):

In their work, Frei et al. describe an approach coupling machine learning algorithms with combinatorial DMS library screening with several mutation per clone. In particular, this approach enabled them to prospectively identify mutations in the RBD domain that could emerge which would maintain affinity for the human ACE2 receptor while potentially evading recognition by existing monoclonal antibodies. The data is mainly used to predict which antibodies or antibody pairs are most likely to be resistant to virus evolution, and thus to identify the antibodies with the broadest and most resilient recognition spectrum. The question of monoclonal antibody resistance to the natural evolution of the SARS-CoV-2 virus is particularly relevant, as many antibodies have been approved for clinical use, but have had to be withdrawn due to their loss of efficacy against new emerging strains.

Overall, I find this work very well done and the methodology very convincing. What the authors call Deep Mutational Learning is an extension of Deep Mutational Scanning taken to a whole new dimension. The technical solution for generating the library is ingenious, and I think it will inspire the scientific community in this field. As explained by the authors, Deep learning becomes essential here to explore the diversity generated and capture the synergistic or combinatorial effects of mutations between them.

I found the logic of the experiments and their explanation very clear (except for one detail, see below) and I strongly recommend the manuscript for publication.

How was the 50 nM ACE2 concentration chosen to sort out the "binder" RBD variants? Is the affinity of the selected clones for ACE2 sufficient to consider that strains expressing such RBD mutants are infectious?

Figure 4. panel e. My understanding of the "Total re-capture" is that the higher the better. Is this the case? This could be indicated in the legend.

Ten sets of 250,000 synthetic sequences are mutated according to the natural mutation frequencies observed for the virus in 2021 or 2022. What is the final spatial distribution of mutations? Is this biased towards the area of interaction with ACE2, the receptor-binding motif? Possibly, is there a risk of introducing

a bias in favour of antibodies whose epitopes outside of the RBM could be under evolutionary constraint?

Why are antibodies used in the form of IgG, which favours avidity effects in the context of Yeast Surface Display? Has consideration been given to working in Fab format, for example, to allow access to more subtle variations than binder/non-binder? Or was it deliberate?

I haven't seen any consideration of the completion of the libraries generated as a function of the mutation rate. In essence, the DMS allows each variant to be sampled many times because the possibilities are very limited. Here, in combination, it is much more difficult to answer this question.

Have the authors assessed the theoretical diversity involved?

Incidentally, how was set the target mutation rate in the libraries?

How resilient are the trained models and the dataset obtained in a BA.1 context with respect to recent strains? Is there a risk that the appearance of strains incorporating an extremely high number of mutations such as BA.2.75 or BA.2.86, could render the dataset obsolete?

I'm a bit puzzled by the results for antibodies 2-7 and COV2-2196. The authors mention a high model uncertainty (lines 300-302). However, the raw data and the "representative FACS dot-plot" of the RBD libraries in the supplementary figure 2 seem to me to indicate a very clear difference between these antibodies and the 4 other antibodies in the same figure. The percentage of cells present in the sorting gates is very low for these two antibodies. Perhaps there is an experimental reason for this difference. Additionally, reference 39 cited by the authors also indicates a very poor affinity of these two antibodies for the BA.1 antigen (KDs of 278 and 508 nM respectively in ref 39) whereas they are better for BA.2 (1.8 and 150 nM). It may be that the choice of these two antibodies in the BA.1 context was not optimal given their low affinity. In any case, the authors could probably discuss this point in more detail, as it seems to me to be less convincing than the rest of the study.

Reviewer #2 (Report for the authors (Required)):

The authors present a deep learning-guided approach to identify antibodies with enhanced resistance to SARS-CoV-2 evolution. This approach is able to predict binding or escape for therapeutic antibodies candidates against diverse RBDs. Therefore, the model can potentially be used for designs the next-generation of antibodies for future variants.

Strengths:

- The method for generating the multisite libraries is novel and will be of use to many others for related and unrelated studies.
- The libraries used for model training cover the most common and latest variants of concern (VOC), including BA.1, BA.2, BA.4, and even XBB.
- The authors developed their models in a comprehensive way, considering many common models and testing their relative performance.
- The analysis of antibody mixtures – and showing how their models can be used to select effective antibody mixtures – is another strength of this paper.

Major weaknesses:

- There is no overview figure. This could help readers know better of the entire paper.
- The paper is written in a somewhat too detailed manner and needs to be broadened. For example, ZCB11 plays an important role in the paper but is not mentioned in the introduction. There are several cases where it feels like this paper has been written for experts and it is not readily accessible to a broader audience. This can be relatively easily fixed, but it needs to be fixed.
- The text in the figures is too small and it is hard to understand the figures as is.

- Figure 3B is important but it is almost impossible to figure out what data is for ACE2 and what is for antibodies. This needs to be much clearer, which requires more than simply increasing text size.
- The authors use different probability cutoffs for their binding and escape, which does not seem justified. If different cutoffs are going to be used, the same cutoffs should also be used and directly compared to the different cutoffs to give a better understanding of the implications of these decisions. As is, it leaves the reader wondering if the authors are tweaking the probability cutoffs to make their models work better.
- When the authors choose CNN as the best model among other machine learning models, it lacks a clear benchmarking for these models. Although they have some comparisons in their Supplementary Figure 7, these are not clear enough why they choose CNN. A table which directly compares each model performance would help readers better understand the reason why they make the decision.
- In Figure 4D, it is not clear if this representation is the best way to evaluate breadth of neutralization. Are there other ways this could be visualized and evaluated?
- In Figure 4E, the concept of “total re-capture” is not clear. Could this be presented and described in a clearer way accessible to a broad audience?

Minor weaknesses:

- In Figure 4F, the term “positional mutation” is misleading. Perhaps this could be changed to “Mutated site in BA.2.86”
- There should be a space between line 357 and 358.
- In line 494, f1 should be F1.
- In 279, the title is not left-aligned with the following paragraph.
- In the first paragraph of Methods section (line 395-407), the font is not consistent. For example, when the protocol is described (line 404-405), the numbers (temperature) seem using Calibri, but others using Times New Roman.

Reviewer #3 (Report for the authors (Required)):

The author constructed a high mutational library on the full-length RBD of Omicron BA.1 and experimentally screened for binding to the ACE2 receptor or neutralizing antibodies. Then the deep learning models were constructed to predict binding or escape of therapeutic antibody candidates. Finally the antibody breadth was assessed by predicting binding or escape to synthetic SARS-CoV-2 variants. The dataset generated is informative and useful to this community. Here are some minor issues.

1. The full datasets are recommended to be submitted into zenodo as only a small subset of the data is available from the github link.
2. For the model comparasion, we recommend to test using pretrained language model to embed the input sequences instead of the one-hot encoding.
3. The author perform class balancing by random subsampling from the majority class to get a equal number of training data of the minority class. This can be a huge waste of training data. The upsampling of minority class is recommended to test.
4. "For a given RBD sequence, each model assigns a binding label if output $P > 0.75$, escape if output $P < 0.25$, or uncertain otherwise. " How did you specify the threshold value?
5. The operation of dilation is more suitable for the the inductive bias of image recognition but not the protein sequences. The hyperparameter of no dilation is recommended to test. And I can not find the final hyperparameter of CNN you chose.

Fri 09 Aug 2024

Decision on Article NBME-23-2751A

Dear Prof Reddy,

Thank you for your patience in waiting for the feedback on your revised manuscript, "Deep learning-guided selection of antibody therapies with enhanced resistance to current and prospective SARS-CoV-2 Omicron variants". Having consulted with the original reviewers (whose comments you will find at the end of this message), I am pleased to write that we shall be happy to publish the manuscript in *Nature Biomedical Engineering*, provided that the points specified in the attached instructions file are addressed.

When you are ready to submit the final version of your manuscript, please upload the files specified in the instructions file.

We encourage authors to take up transparent peer review. If you are eligible and opt in to transparent peer review, we will publish, as a single supplementary file, all the reviewer comments for all the versions of the manuscript, your rebuttal letters, and the editorial decision letters. **If you opt in to transparent peer review, in the attached file please tick the box 'I wish to participate in transparent peer review'; if you prefer not to, please tick 'I do NOT wish to participate in transparent peer review'**. In the interest of confidentiality, we allow redactions to the rebuttal letters and to the reviewer comments. If you are concerned about the release of confidential data, please indicate what specific information you would like to have removed; we cannot incorporate redactions for any other reasons. More information on transparent peer review is available.

Best wishes,

Pep

Pep Pàmies
Chief Editor, Nature Biomedical Engineering

P.S. Nature Portfolio journals encourage authors to share their step-by-step experimental protocols on a protocol-sharing platform of their choice. Nature Portfolio's Protocol Exchange is a free-to-use and open resource for protocols; protocols deposited in Protocol Exchange are citable and can be linked from the published article. More details can be found at www.nature.com/protocolexchange/about.

Reviewer #1 (Report for the authors (Required)):

The changes made by the authors address the concerns and comments raised during the first stage of the review.

I think the revisions have also made the article clearer. The addition of several paragraphs not only justifies certain aspects, but also outlines possible future developments. In additions, the changes make it easier to interpret certain graphics.

I have no further questions regarding this manuscript.

Reviewer #2 (Report for the authors (Required)):

The authors have generally addressed my concerns. Thank you!

A few remaining minor points:

1. Table 1 should be in the supplement. It is too specific for a general audience.
2. Figure 2 font in part A is still too small. Either make this clearer or remove it, as it is ineffective as is.

Reviewer #3 (Report for the authors (Required)):

In this revision, the authors have adequately addressed the issues raised in the previous round of review. I feel that the manuscript is appropriate for publication in its current form.

Nature Biomedical Engineering is a Transformative Journal. Authors may publish their research with us through the traditional subscription access route, or make their paper immediately open access through payment of an article-processing charge. More information about publication options is available.

You may need to take specific actions to comply with funder and institutional open-access mandates. If the work described in the accepted manuscript is supported by a funder that requires immediate open access (as outlined, for example, by Plan S) and your manuscript was originally submitted on or after January 1st 2021, then you will need to select the gold OA route. Authors selecting subscription publication will need to accept our standard licensing terms (including our self-archiving policies), and these will supersede any other terms that the author or any third party may assert apply to any version of the manuscript.

Rebuttal 1

Response to Reviewers

Reviewer #1:

1. **How was the 50 nM ACE2 concentration chosen to sort out the "binder" RBD variants? Is the affinity of the selected clones for ACE2 sufficient to consider that strains expressing such RBD mutants are infectious?**

50 nM ACE2 has also been used in other studies using yeast displayed RBD (Taft, Weber et al., Cell, 2022). Affinities of naturally occurring RBD variants for ACE2 are in the single to double digit nM range, for example Wu-Hu-1 (27.5 nM), Alpha (6.7 nM), Beta (19.7 nM), Gamma (16.0 nM), Delta (25.1 nM) or Omicron (5.5 nM) (Li et al., Cell, 2022; Kim et al., J. Comput. Chem., 2023). It is therefore expected that the affinities of the yeast-display RBD variants that can be detected as binders to 50 nM ACE2 are in a physiological range consistent with infectious SARS-CoV-2 variants .

We added this explanation to the revised method section on page 18, paragraph 2.

2. **Figure 4. panel e. My understanding of the "Total re-capture" is that the higher the better. Is this the case? This could be indicated in the legend.**

Yes, this is the correct interpretation of "Total re-capture", we have thus incorporated this suggestion into the revised Figure legend.

3. **Ten sets of 250,000 synthetic sequences are mutated according to the natural mutation frequencies observed for the virus in 2021 or 2022. What is the final spatial distribution of mutations? Is this biased towards the area of interaction with ACE2, the receptor-binding motif? Possibly, is there a risk of introducing a bias in favour of antibodies whose epitopes outside of the RBM could be under evolutionary constraint?**

We have now included our final generated sequences in the dataset that is deposited on Zenodo (<https://zenodo.org/records/11172179>).

To answer the question specifically, yes the final distribution of mutations are focused on certain regions of the spike protein, including the RBM. It is difficult to determine for certain if generating sequences this way creates an artificial bias in favour of antibodies with epitopes outside the RBM. Studies using long-term evolutionary data of SARS-CoV-2 have identified mutations in the RBD to be primary drivers of the virus' evolution - correlated with increased ACE2 binding, immune evasion, and overall viral fitness (Gayvert et al. 2023; Ma et al. 2023). Furthermore, the RBM is particularly immunodominant and thus is subject to even greater immune selection pressures. Antibody accessibility (von Bülow et al. 2023), as well as models trained using antibody escape DMS data (Wang et al. 2023; Han et al. 2023) have been used to successfully pinpoint and forecast the RBM as a site of future mutational activity. Since those regions are under greater evolutionary constraint, antibodies targeting those regions will naturally have greater "breadth" in the evolutionary landscape since the virus is unlikely to mutate those epitopes and escape. Thus generating more variants with mutations outside of clearly defined hotspots would in fact cause us to underreport the breadth of those antibodies that target more conserved regions in the RBD and also would be farther away from the natural evolutionary trends observed in GISAID data.

4. **Why are antibodies used in the form of IgG, which favours avidity effects in the context of Yeast Surface Display? Has consideration been given to working in Fab format, for example, to allow access to more subtle variations than binder/non-binder? Or was it deliberate?**

Antibody therapies for COVID-19 have thus far all been in the IgG format. To profile therapeutic antibody candidates, we considered it important to maintain this format.

While Fab formats would reduce potential avidity effects, they may result in a higher fraction of RBD variants that appear to escape (non-binding to Fab), but may still have partial binding in an IgG format. During the pandemic, the antibody therapy from Astra Zeneca (Evusheld or AZD8895+AZD1061) had a substantial loss of binding to Omicron variants (Cao et al. 2022), but was still approved for clinical use with an adjustment in dosing

(see: **update from FDA 02/24/2022**: <https://www.fda.gov/drugs/drug-safety-and-availability/fda-announces-evusheld-not-currently-authorized-emergency-use-us>).

Therefore, it is now established that even antibodies with a substantial loss in affinity can still receive clinical authorization and use, therefore screening with the higher avidity IgG format allows us to identify only complete escape variants.

- 5. I haven't seen any consideration of the completion of the libraries generated as a function of the mutation rate. In essence, the DMS allows each variant to be sampled many times because the possibilities are very limited. Here, in combination, it is much more difficult to answer this question. Have the authors assessed the theoretical diversity involved?**

Considering all possible combinations of fragments with either 0,1 or 2 mutations for each seq-library, the theoretical diversity is calculated to be approximately 10^{42} . From this space, we sampled 2×10^8 variants. Although the theoretical space has been undersampled, recent work has shown that when combining deep sequencing data from high-throughput selections, even from an undersampled protein sequence space, supervised machine learning models are able to classify and accurately predict binding or non-binding on large (unseen) parts of this protein sequence space (Minot and Reddy, Cell Systems, 2024; Taft et al., Cell, 2022; Mason et al., Nature Biomed Eng, 2021).

To address this question, we included additional information in our revised manuscript on page 4.

- 6. Incidentally, how was set the target mutation rate in the libraries?**

By pooling fragments with either 0, 1 or 2 mutations in different ratios, we were able to obtain libraries with different mutational distributions. The revised Fig. 2e now shows the mutational distributions of these different libraries.

Of the tested mutational distributions, we picked the library with the highest average number of mutations. This particular library allowed us to profile and model highly mutated Omicron variants, while still capturing low distance variants.

We included this information in the revised Results section on page 6, paragraph 2.

- 7. How resilient are the trained models and the dataset obtained in a BA.1 context with respect to recent strains? Is there a risk that the appearance of strains incorporating an extremely high number of mutations such as BA.2.75 or BA.2.86, could render the dataset obsolete?**

Based on the data we present in the revised Fig. 4c, we can conclude that our models perform well on extensively mutated Omicron variants BQ.1 and XBB.1. In addition, in Fig. 5f, we show that the models are able to make reliable predictions on highly mutated variants such as BA.2.86: we identify positions and/or mutations with high escape scores that are present in BA.2.86. We believe this is due to the models' ability to learn and infer the combinatorial effects between mutations based on the design of the mutagenesis libraries.

Although there will be a limit to the predictive performance of these deep learning models, given that there are complex epistatic effects between mutations across the spike protein, our combinatorial libraries provides greater coverage of the mutational sequence space of the RBD compared to traditional DMS based methods (Starr et al. 2020; Starr et al. 2021), and thus will maintain predictive performance for

longer. Furthermore, our strategy that we describe for combinatorial mutagenesis library design and assembly (Fig. 2) provides modularity and would enable us to update relatively easily with only a partial set of new ssODNs.

8. **I'm a bit puzzled by the results for antibodies 2-7 and COV2-2196. The authors mention a high model uncertainty (lines 300-302). However, the raw data and the "representative FACS dot-plot" of the RBD libraries in the supplementary figure 2 seem to me to indicate a very clear difference between these antibodies and the 4 other antibodies in the same figure. The percentage of cells present in the sorting gates is very low for these two antibodies. Perhaps there is an experimental reason for this difference. Additionally, reference 39 cited by the authors also indicates a very poor affinity of these two antibodies for the BA.1 antigen (KDs of 278 and 508 nM respectively in ref 39) whereas they are better for BA.2 (1.8 and 150 nM). It may be that the choice of these two antibodies in the BA.1 context was not optimal given their low affinity. In any case, the authors could probably discuss this point in more detail, as it seems to me to be less convincing than the rest of the study.**

We thank the reviewer for their comment about the results for antibodies 2-7 and COV2-2196. They are indeed correct about the lower binding of these antibodies, we have now added more text to elaborate on these antibodies on page 8, paragraph 3 :

“During the sorting process, it was noted that antibodies COV2-2196 and 2-7 show a weaker binding signal (Supplementary Fig. 2). This was especially pronounced in the case of antibody 2-7 and is likely due to the low affinity of this antibody to Omicron BA.1 RBD (Supplementary Fig. 10) and a generally low mutational resilience (Supplementary 3a). Those factors contributed to the collection of fewer cells for those antibodies.”

and discuss the impact of affinity on DML predictions on page 17, paragraph 1.

“A current limitation that faces the DML approach is a sensitivity issue when it comes to making predictions for low affinity antibodies. Future work can explore the possibility of sorting at multiple antibody concentrations and building multi-label or regression models to predict quantitative changes in antibody affinity to given variants, rather than a binding/non-binding label obtained from our current classifiers. The resulting predictions would be more nuanced in cases where antibody affinities are already weak, such as antibodies 2-7 and COV2-2196.”

Reviewer #2 (Report for the authors (Required)):

Major weaknesses:

1. **There is no overview figure. This could help readers know better of the entire paper.**

We thank R2 for this comment. On page 3, we have now included a new overview figure (Fig. 1) that describes the overall study.

2. **The paper is written in a somewhat too detailed manner and needs to be broadened. For example, ZCB11 plays an important role in the paper but is not mentioned in the introduction. There are several cases where it feels like this paper has been written for experts and it is not readily accessible to a broader audience. This can be relatively easily fixed, but it needs to be fixed.**

We have added more text introducing ZCB11 and its discovery in the initial introduction. In addition, we have included more detailed descriptions of some of the concepts used within the paper (such as the binding “landscape” and antibody “re-capture”) into the corresponding results sections with the aim to make it more easily accessible and provide more clarity.

3. The text in the figures is too small and it is hard to understand the figures as is.

We appreciate the feedback from R2, we have updated many of the main figures with larger panels and text sizes for improved readability.

Specifically, the following changes to Main Figures:

- Fig. 1: A new overview figure has been added
- Fig. 2: We have increased the font size of axis labels in all panels, as well as changing panel e) for greater distinction between the oligo mix methods
- Fig. 3: Panels c-f) have increased font sizes in all axis and legend labels.
- Fig. 4: Overall increase in size of panels. Previously Fig. 3b has been changed into Table 1 to clearly show the performance of the final models. In addition, the new Fig. 4c presents the predictions of models using the new threshold of binding as $P > 0.5$ and escape as $P \leq 0.5$.
- Fig. 5: We altered the sizes of a majority of the panels for visibility. In addition, instead of the previous UMAP, we now depict the ZCB11 escape variants in Fig. 5d as a phylogenetic tree to highlight the complementary coverage of three antibodies (A23-58.1, ADG20, and Brii-198). Furthermore, all panels have been updated to represent results using the new updated thresholds of binding as $P > 0.5$ and escape as $P \leq 0.5$.

Additional changes to Supplementary Figures:

- Supplementary Fig. 10 has been updated with improved titration curves
- Supplementary Table 6 now includes the final parameters used to train our final CNN models.
- Previous supplementary figure 7 has now been changed to Supplementary Table 5, which clearly shows all performance metrics of our initial baseline models trained on our library datasets.

4. Figure 3B is important but it is almost impossible to figure out what data is for ACE2 and what is for antibodies. This needs to be much clearer, which requires more than simply increasing text size.

We understand the R2's concern and to provide the data in a manner that is easier to interpret, we replaced Fig. 3B (now Fig. 4 in revision) with a new Table 1 that shows the final performance metrics of all final models trained on ACE2 and antibodies.

See Table 1 on page 14.

5. The authors use different probability cutoffs for their binding and escape, which does not seem justified. If different cutoffs are going to be used, the same cutoffs should also be used and directly compared to the different cutoffs to give a better understanding of the implications of these decisions. As is, it leaves the reader wondering if the authors are tweaking the probability cutoffs to make their models work better.

Both R2 and R3 have pointed out the use of different probability cutoffs for binding and escape in our machine learning models. The selection of these cutoffs were motivated by our previous publication (Taft et al. 2022), in which we explored more stringent cutoff values to make more conservative predictions.

In response to the reviewer's suggestions, we have re-run all the analysis using a standard threshold that is very easy to interpret and justify: $P > 0.5$ is classified as binding, and $P \leq 0.5$ is classified as escape. Using these standard thresholds, we now report for our validation set (natural variants of Omicron sublineages) that our predictions are correct for 87.5% (63/72) of ACE2-RBD variants or antibody-RBD variants (see revised Fig. 4c).

With these standard thresholds that are justified and easy to interpret, there were no major changes observed in the results of our synthetic lineage predictions, as well as breadth and escape score calculations (see revised Fig. 5). We have updated the text to reflect this, please see page 11, paragraph 2.

6. **When the authors choose CNN as the best model among other machine learning models, it lacks a clear benchmarking for these models. Although they have some comparisons in their Supplementary Figure 7, these are not clear enough why they choose CNN. A table which directly compares each model performance would help readers better understand the reason why they make the decision.**

See also response to R2 comment 3.

We thank R2 for pointing this out, to address this, we now provide a new Table 1 and new Supplementary Table 6, which provide performance metrics (MCC) for all final and benchmarking models, respectively.

Some baseline XGBoost models achieved better performance than the CNN models in particular cases. The justification for still using CNN models is described in the revised manuscript (see page 10, paragraph 2).

7. **In Figure 4D, it is not clear if this representation is the best way to evaluate breadth of neutralization. Are there other ways this could be visualized and evaluated? In Figure 4E, the concept of “total re-capture” is not clear. Could this be presented and described in a clearer way accessible to a broad audience?**

We thank the reviewer for this very useful feedback about visualization and interpretation. To address this, we have generated a new Fig. 5d (previously Fig. 4d), which presents a phylogenetic tree to visualize synthetic lineages of escape variants to ZCB11, and the binding of the three other antibodies providing complementary breadth across the tree nodes. We believe this visualization should provide more clarity into the complementary nature of binding between combinations of antibodies across synthetic mutational trajectories. Please also see the revised and associated text in the Results section (see page 14, paragraph 1).

Furthermore, to address the concept of “re-capture”, we added a more in-depth description and discussion to our Results section (page 19, paragraph 2) where we explicitly define the concept of “re-capture”.

Minor weaknesses:

- **In Figure 4F, the term “positional mutation” is misleading. Perhaps this could be changed to “Mutated site in BA.2.86”**

We corrected this.

- **There should be a space between line 357 and 358.**

We corrected this.

- **In line 494, f1 should be F1.**

We corrected this.

- **In 279, the title is not left-aligned with the following paragraph.**

We corrected this.

- **In the first paragraph of Methods section (line 395-407), the font is not consistent. For example, when the protocol is described (line 404-405), the numbers (temperature) seem using Calibri, but others using Times New Roman.**

We corrected this.

Reviewer #3 (Report for the authors (Required)):

1. The full datasets are recommended to be submitted into zenodo as only a small subset of the data is available from the github link.

We have now uploaded the datasets to Zenodo at <https://zenodo.org/records/11172179> (doi: [10.1101/2023.10.09.561492](https://doi.org/10.1101/2023.10.09.561492))

2. For the model comparasion, we recommend to test using pretrained language model to embed the input sequences instead of the one-hot encoding.

We thank R3 for this valid suggestion. We tested the performance of using embeddings from a mid-sized ESM model (esm2_t33_650M_UR50D) by training Support Vector Machine (SVM) and XGBoost classifiers. We chose to test with these relatively simpler ML models first, rather than deep learning models CNN/MLPs, because we expect that if ESM-embeddings truly offered improved performance, we should observe performance improvements of such simple ML models.

We attach below a table that shows MCC scores of all of our baseline models (as in new Supplementary Table 6) in addition to the ESM models (highlighted in light orange). We observed a marked decrease in model performance when using Base-ESM embeddings compared to one-hot encoding. We also experimented with fine-tuning the ESM model on our dataset by unfreezing the last layer, and training for 50-epochs with a single linear head (Tuned-ESM). Using the embeddings from Tuned-ESM also failed to improve the performance of the models. Our results are consistent with another recent study that showed embeddings from pre-trained protein language models do not always lead to improved performance versus more simple embeddings (one-hot encoding) (Li et al. 2024). Given the design of our libraries and our experiment, it is likely that this may explain the lower than expected performance of models trained with ESM embeddings. Additionally, we are in the process of developing a follow-up study that interrogates the use of protein language models in the context of DML.

Seq-library A									
Model	2-7	A23-58.1	ACE2	ADG20	Brii-198	COV2-2196	S2H97	S2X259	ZCB11
KNN	0.864±0.008	0.771±0.008	0.887±0.002	0.774±0.013	0.943±0.004	0.618±0.013	0.756±0.009	0.564±0.017	0.837±0.015
Naive Bayes	0.85±0.014	0.798±0.011	0.9±0.005	0.874±0.009	0.936±0.007	0.665±0.027	0.816±0.008	0.639±0.007	0.863±0.012
Logistic Regression	0.867±0.008	0.81±0.011	0.934±0.004	0.873±0.01	0.946±0.002	0.69±0.02	0.811±0.005	0.657±0.015	0.873±0.006
Random Forest	0.849±0.01	0.801±0.014	0.893±0.004	0.844±0.015	0.928±0.01	0.701±0.016	0.783±0.015	0.638±0.007	0.831±0.01
XGBoost	0.878±0.009	0.843±0.012	0.96±0.003	0.882±0.009	0.942±0.005	0.723±0.018	0.827±0.007	0.667±0.021	0.878±0.016
SVM	0.885±0.007	0.809±0.019	0.939±0.013	0.873±0.011	0.948±0.006	0.654±0.031	0.784±0.037	0.647±0.029	0.872±0.01
MLP	0.887±0.015	0.804±0.02	0.948±0.003	0.871±0.011	0.951±0.018	0.68±0.019	0.841±0.022	0.676±0.018	0.87±0.009
CNN	0.911±0.017	0.848±0.001	0.946±0.002	0.894±0.003	0.967±0.008	0.653±0.001	0.835±0.021	0.829±0.001	0.869±0.015
Base-ESM+XGBoost	0.696±0.012	0.53±0.009	0.728±0.006	0.724±0.012	0.782±0.012	0.399±0.021	0.574±0.017	0.318±0.021	0.596±0.025
Base-ESM+SVM	0.739±0.014	0.487±0.013	0.779±0.006	0.761±0.009	0.81±0.011	0.387±0.009	0.642±0.016	0.192±0.024	0.623±0.02
Tuned-ESM+XGBoost	0.704±0.013	0.542±0.018	0.73±0.004	0.717±0.007	0.786±0.01	0.377±0.014	0.569±0.016	0.32±0.024	0.604±0.019
Tuned-ESM+SVM	0.754±0.023	0.477±0.012	0.789±0.005	0.76±0.009	0.826±0.014	0.35±0.038	0.617±0.006	0.226±0.022	0.606±0.009

Seq-library B									
Model	2-7	A23-58.1	ACE2	ADG20	Brii-198	COV2-2196	S2H97	S2X259	ZCB11
KNN	0.815±0.006	0.937±0.004	0.927±0.002	0.916±0.006	0.888±0.004	0.66±0.006	0.933±0.004	0.646±0.009	0.96±0.002
Naive Bayes	0.841±0.006	0.963±0.004	0.938±0.002	0.947±0.004	0.907±0.008	0.709±0.023	0.945±0.004	0.708±0.012	0.967±0.002
Logistic Regression	0.864±0.004	0.977±0.003	0.963±0.001	0.944±0.005	0.908±0.005	0.723±0.019	0.955±0.003	0.737±0.013	0.974±0.003
Random Forest	0.865±0.008	0.973±0.002	0.919±0.002	0.935±0.006	0.881±0.005	0.71±0.01	0.953±0.002	0.722±0.014	0.966±0.003
XGBoost	0.908±0.006	0.975±0.002	0.971±0.002	0.952±0.003	0.928±0.007	0.798±0.01	0.955±0.004	0.787±0.013	0.973±0.003
SVM	0.863±0.01	0.98±0.003	0.962±0.003	0.953±0.004	0.915±0.005	0.709±0.019	0.959±0.008	0.709±0.014	0.976±0.003
MLP	0.875±0.011	0.977±0.002	0.964±0.011	0.951±0.01	0.917±0.011	0.747±0.019	0.96±0.005	0.727±0.039	0.974±0.01
CNN	0.884±0.004	0.983±0.002	0.962±0.004	0.959±0.001	0.926±0.005	0.791±0.006	0.965±0.002	0.74±0.022	0.977±0.002
Base-ESM+XGBoost	0.631±0.004	0.885±0.006	0.797±0.004	0.817±0.006	0.765±0.012	0.435±0.015	0.82±0.008	0.415±0.01	0.868±0.005
Base-ESM+SVM	0.651±0.009	0.918±0.005	0.81±0.005	0.87±0.005	0.762±0.011	0.416±0.022	0.844±0.007	0.366±0.007	0.887±0.005
Tuned-ESM+XGBoost	0.635±0.015	0.889±0.003	0.799±0.006	0.816±0.004	0.764±0.008	0.427±0.01	0.819±0.006	0.424±0.017	0.867±0.003
Tuned-ESM+SVM	0.653±0.013	0.916±0.004	0.817±0.005	0.865±0.004	0.768±0.007	0.405±0.012	0.852±0.007	0.37±0.026	0.889±0.004

3. The author perform class balancing by random subsampling from the majority class to get a equal number of training data of the minority class. This can be a huge waste of training data. The upsampling of minority class is recommended to test.

We thank R3 for this valuable suggestion. We have updated the baseline model performances with new hyperparameter optimized models, along with an additional baseline model (XGBoost). These models were trained with upsampling of the minority class and their performance is now described in the new Supplementary Table 5.

In this process, we also noticed that the optimized XGBoost model actually shows a slightly better performance than the final CNN model. However, for one particular condition: seq-library A of S2X259, the CNN models vastly outperform any other model. This is most likely due to long-distance interactions that were learned across the length of the RBD. Since mutations can occur that span across the length of the entire 201 a.a. RBD, it is still preferable to use a deep learning approach such as a CNN model.

We have updated the manuscript to contain a more in-depth discussion of these results: see page 11, paragraph 1.

4. "For a given RBD sequence, each model assigns a binding label if output $P > 0.75$, escape if output $P < 0.25$, or uncertain otherwise." How did you specify the threshold value?

This comment is similar to R2, comment 5. Therefore we provide below the same response.

Both R2 and R3 have pointed out the use of different probability cutoffs for binding and escape in our machine learning models. The selection of these cutoffs were motivated by our previous publication (Taft et al. 2022), in which we explored more stringent cutoff values to make more conservative predictions.

In response to the reviewer's suggestions, we have re-run all the analysis using a standard threshold that is very easy to interpret and justify: $P > 0.5$ is classified as binding, and $P \leq 0.5$ is classified as escape. Using these standard thresholds, we now report for our validation set (natural variants of Omicron sublineages) that our predictions are correct for 87.5% (63/72) of ACE2-RBD variants or antibody-RBD variants (see revised Fig. 4c).

With these standard thresholds that are justified and easy to interpret, there were no major changes observed in the results of our synthetic lineage predictions, as well as breadth and escape score calculations (see revised Fig. 5). We have updated the text to reflect this, please see page 11, paragraph 2.

5. The operation of dilation is more suitable for the the inductive bias of image recognition but not the protein sequences. The hyperparameter of no dilation is recommended to test. And I can not find the final hyperparameter of CNN you chose.

We appreciate R3's suggestion regarding dilation and hyperparameters. While dilations are indeed often used in image-based learning tasks, we were inspired by the previously published ProtCNN model (Bileschi et al. 2022), which combined dilations with a ResNet architecture and thus it considers both local and global information for protein engineering tasks. Since the RBD mutagenesis libraries span across the 201 a.a. length of the RBD, we thought it appropriate to test out this architecture for our applications. We include below a table depicting the MCC scores of two baseline CNN models we initially trained during the benchmarking stages using our initial default hyperparameters, one with a dilation of 0 (cnn-dil0), and another with a dilation of 2 (cnn-dil2), along with the scores of our final models used in the rest of our study (cnn_final).

Seq-library A									
Model	2-7	A23-58.1	ACE2	ADG20	Brii-198	COV2-2196	S2H97	S2X259	ZCB11
cnn_dil0	0.869±0.038	0.811±0.014	0.943±0.005	0.88±0.008	0.956±0.008	0.615±0.017	0.82±0.021	0.812±0.011	0.847±0.045
cnn_dil2	0.907±0.008	0.818±0.011	0.945±0.003	0.877±0.007	0.949±0.008	0.626±0.015	0.821±0.023	0.809±0.028	0.861±0.009
cnn_final	0.911±0.017	0.848±0.001	0.946±0.002	0.894±0.003	0.967±0.008	0.653±0.001	0.835±0.021	0.829±0.001	0.869±0.015

Seq-library B									
Model	2-7	A23-58.1	ACE2	ADG20	Brii-198	COV2-2196	S2H97	S2X259	ZCB11
cnn_dil0	0.891±0.013	0.976±0.003	0.958±0.004	0.951±0.005	0.916±0.009	0.793±0.008	0.95±0.006	0.742±0.009	0.972±0.006
cnn_dil2	0.887±0.015	0.974±0.005	0.956±0.012	0.952±0.005	0.903±0.02	0.782±0.007	0.953±0.008	0.73±0.022	0.974±0.003
cnn_final	0.884±0.004	0.983±0.002	0.962±0.004	0.959±0.001	0.926±0.005	0.791±0.006	0.965±0.002	0.74±0.022	0.977±0.002

From the benchmarking results, we found that with default parameters, the use of a dilation rate = 2 led to improvements in MCC scores across some of our antibody datasets. In particular, we were drawn to the result seen in seq-library A for antibody 2-7, where we observed a 0.04 increase in MCC performance. We believe this indicated there was a benefit to learning global patterns across the RBD with the use of dilations, and was substantial enough to pursue using dilations during hyperparameter optimization. Additionally, the majority of our final models showed further improved performance over these baselines, and, in cases where the initial 0-dilation models outperformed 2-dilation, the final models achieved equal or exceeded the performance of baseline models.

We have updated the manuscript with a new Supplementary Table 6 that lists the hyperparameters used for the final CNN models.